# TableFactory: Generating Semantically Linked Tabular Data via Multi-Agent Behavioral Simulation

**Mingxuan Liu** [1]  **Xiangjian Jiang** [2]  **Johannes Hoffart** [1]  **Tassilo Klein** [1]

## Abstract

Multi-table operational data is scarce, and much synthetic tabular work targets single-table distribution matching. We present **TableFactory**, a prompt-conditioned multi-agent simulator that produces semantically linked relational tables for small-business scenarios. From a short text prompt (e.g., *"simulate a butcher shop in a morning market in Barcelona"*), an LLM-backed generator builds a complete business configuration, such as catalogs, suppliers, staffing, demand, and policies, augmented with macro attributes such as GDP and tax rates. Five abstract entity roles interact through a shared relational world state, materializing a fixed schema of twenty foreign-key-linked tables per run. The environment supports LLM, RL (PPO, SAC, DDPG), and rule-based controllers, together with an MBTI-style persona module for behavioral diversity. We report diagnostic studies of controller interchangeability, scenario and regional variation, and persona effects, and release reproducible table bundles with configurations and metadata. Code is open-sourced.

## 1. Introduction

Tabular data underpins decision-making across enterprise operations, healthcare, finance, education, and transportation. As models for tabular understanding and structured reasoning continue to advance, further progress depends on large-scale, diverse, and well-documented corpora. In practice, however, high-value structured data is often private, fragmented, or tightly coupled to operational workflows, while accessible datasets frequently lack scale, metadata, or multi-table context. This gap is especially pronounced in everyday business settings, where records are distributed across tables linked by foreign keys rather than stored in a single-table dataset. For example, in an electronics store, an order may link to a payment record, a shipment entry, a return request, and a customer support ticket.

Synthetic tabular data generation is a natural response to the scarcity and privacy sensitivity of real structured data. Existing approaches include classical statistical and machine-learning generators (Sklar, 1973; Wang et al., 2023), neural methods such as VAEs, GANs, and diffusion models (Park et al., 2018; Liu et al., 2023; Kotelnikov et al., 2023; Ceritli et al., 2023; Zhang et al., 2023), and more recent LLM-based methods (Kim et al., 2024; Zhao et al., 2025b). Most of this literature, however, focuses on *approximating the distribution* of an observed table, or at most a small fixed set of tables, typically under a single-table assumption. Real operational tabular data is different: it is generated by *behavioral processes* in which customers browse and buy, staff handle tickets and work orders, managers approve exceptions, and inventory is replenished over time. As a result, consistency must hold not only within rows, but also across entities, timestamps, and foreign-key-linked tables (Klein et al., 2025; Spinaci et al., 2025). Yet such enterprise data remains scarce and difficult to scale under the same privacy and access constraints (Mulang et al., 2026).

A complementary perspective is to simulate the processes that produce rows, so that cross-table consistency arises from shared state and interaction rather than post-hoc coupling of independent generators. Related simulation ideas have shown promise in multi-agent environments driven by LLMs or RL agents, including urban life (Yang et al., 2024; Yu et al., 2026), competitive restaurant settings (Zhao et al., 2023), macroeconomic activities (Li et al., 2024; Mi et al., 2025), and trading (Zhao et al., 2025a); Anthropic (2025) also piloted an automated small office store operated by Claude Sonnet. Motivated by these advances, we ask: "*Instead of approximating table distributions directly, can we simulate the underlying behavioral processes so that semantically linked multi-table data emerges naturally as a byproduct?*"

We introduce **TableFactory**, a synthetic data generation framework that produces semantically linked relational tables through *multi-agent behavioral simulation*. Given a one-sentence business description in natural language,

---

[1]SAP SE [2]University of Cambridge. Correspondence to: Mingxuan Liu <mingxuan.liu01@sap.com>.

*Proceedings of the 2nd ICML Workshop on Foundation Models for Structured Data*, Seoul, South Korea. 2026. Copyright 2026 by the author(s).

*Table 1.* **Comparison of tabular data synthesis paradigms.** Existing methods mainly generate data by approximating observed table distributions, whereas our behavioral simulation generates data by simulating the underlying processes that produce the tables.

| Paradigm | Modeling Target | Mechanism | Cross-domain | User Control | Output |
|---|---|---|---|---|---|
| Traditional Generation | Table distribution | Statistical / VAE / GAN | ✗ | ✗ | Single table |
| Diffusion Model | Table distribution | Diffusion-based generation | ✗ | ✗ | Single table |
| LLM-based | Table distribution | LLM Prompting / fine-tuning | ✗ | ✗ | Single table |
| **Behavioral Simulation (Ours)** | Behavioral process | Multi-agent (LLM / RL) simulation | ✓ | ✓ | Multiple linked tables |

TableFactory automatically constructs a scenario with products, suppliers, staffing, demand patterns, and policies by combining the world knowledge of LLMs with web-sourced economic grounding. It then instantiates five domain-agnostic entities, Customer, Sales, Operations, Inventory, and Manager, each played by an LLM or RL agent with diverse sampled personas (*e.g.*, grumpy or patient), interacting through a shared relational world state. The simulation exports linked tables spanning orders, payments, shipments, support tickets, work orders, and inventory records. We focus on small street businesses such as cafes, boutiques, and food retail as a concrete setting where operational structure, cross-table consistency, and diverse customer behavior naturally co-occur. Tab. 1 compares our approach with prior tabular data synthesis methods.

To our knowledge, TableFactory is the first tabular data synthesis framework based on *multi-agent behavioral simulation* rather than single-table distribution approximation. Our contributions are: *i)* a pipeline that maps a natural-language business description to semantically linked relational tables through simulated operations; *ii)* a unified architecture with abstract entity roles, interchangeable controllers (LLM or RL), and a shared relational world state; and *iii)* diagnostic evidence of cross-scenario flexibility and controllable behavioral diversity, with high-fidelity tabular data.

## 2. Method

**System workflow and overview.** TableFactory converts a free-form business description into semantically linked relational tables through multi-agent behavioral simulation (Fig. 1). The user provides a business description, selects an agent backend (LLM or RL), and specifies the simulation length. The pipeline has four stages. A **scenario generation layer** uses an LLM to produce a business configuration (products, suppliers, staffing, demand, policies). An **entity layer** instantiates five abstract business roles and specializes them via the generated configuration. An **agent layer** assigns each role a controller (LLM, RL, or rule-based), while a **persona layer** injects behavioral traits for diversity. Finally, a **runner** executes the simulation over multiple days: entities observe a shared relational world state, agents act, and business events update the state. The resulting operational logs are exported as foreign-key-linked CSV tables with run metadata.

### 2.1. Scenario Generation Layer

The scenario generation layer bridges natural-language input and executable simulation. Given a free-form prompt, TableFactory first extracts location information, such as city, country, and currency, and augments it with lightweight macroeconomic attributes, such as GDP and corporate tax rate, sourced from the internet (Trading Economics, 2026) available. An LLM then generates a structured business configuration containing the product catalog, suppliers, staffing, payment methods, store policies, demand patterns, and role descriptions. For example, from a butcher-shop prompt in Barcelona, the system extracts Barcelona, Spain, and EUR, and may further ground the scenario with country-level attributes such as GDP and tax rate. The LLM then generates a localized product catalog, regional suppliers, a morning-weighted demand model with weekend variation, and a matching staffing plan. This defines both the static business state and the operating conditions of the simulation.

The pipeline is then specialized to the chosen agent backend. For LLM agents, TableFactory generates role-specific system prompts that ground each entity in the scenario. For RL agents, it generates scenario-specific reward weights that reflect business priorities, such as stockout penalties or service efficiency. This prompt-conditioned design also enables scenario and regional variation: changing the business description or location changes the operating context while preserving the same simulation engine.

### 2.2. Entity Layer

The entity layer encodes business logic through five abstract roles, referred to as *entities* throughout this paper: *Customer*, *Sales*, *Operations*, *Inventory*, and *Manager*. These roles are domain-agnostic and specialized by the generated configuration rather than by business-specific hardcoding. For example, in the Barcelona butcher-shop scenario, Operations acts as the butcher responsible for meat preparation and food-safety handling, Sales serves customers and recommends cuts, Inventory monitors cold storage and perishability, and Manager coordinates suppliers and approves higher-value refunds. In other scenarios, the same roles adapt naturally: Operations may represent a bartender in a cafe or product handling staff in a boutique.

During simulation, each entity reads from and writes to a shared relational world state, an in-memory store covering

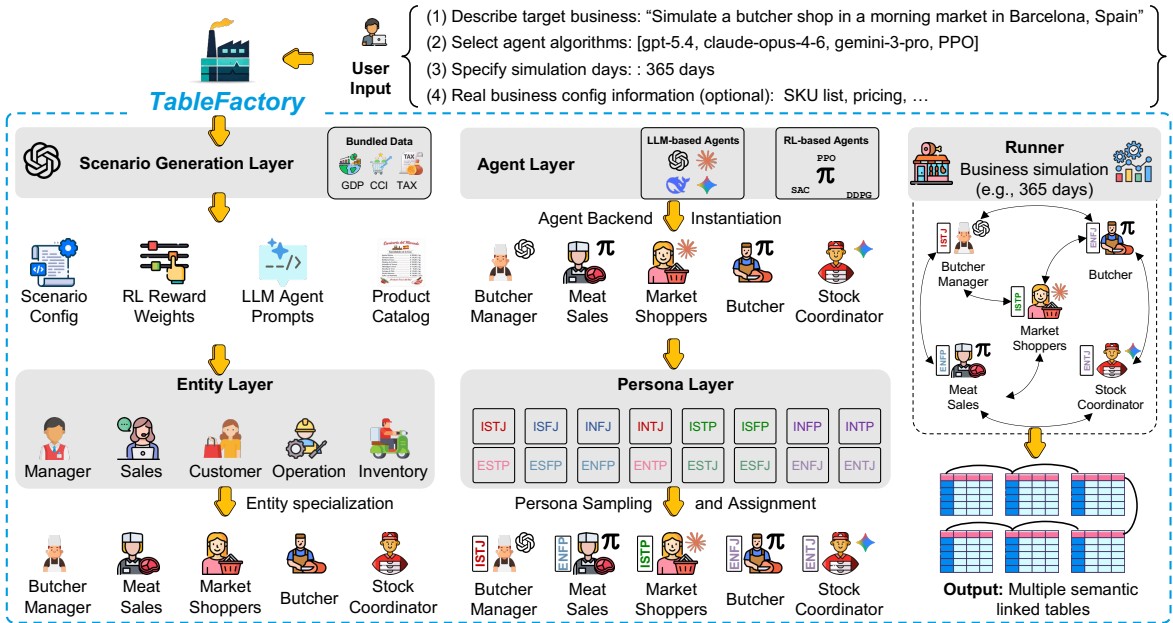

*Figure 1.* **System overview and user-side pipeline of TableFactory.**

customers, products, carts, orders, payments, support tickets, work orders, purchase orders, inventory updates, and staff actions. Entities do not communicate directly; instead, interactions emerge through state changes. In the butcher-shop example, a customer order reduces inventory and may trigger replenishment, a product requiring preparation creates a work order for Operations, and a complaint creates a support ticket for Sales. This shared-state design is central to TableFactory, as cross-table consistency arises from simulated process interactions rather than post-hoc table alignment.

### 2.3. Agent Layer

The agent layer determines how each entity is controlled during simulation. TableFactory supports three agent families: LLM-based, reinforcement learning (RL), and rule-based. LLM agents (Achiam et al., 2023; Team et al., 2023) receive role-specific prompts and current observations, then return structured decisions. RL agents map numerical observations to actions through learned policies, with PPO (Schulman et al., 2017), SAC (Haarnoja et al., 2018), and DDPG (Lillicrap et al., 2016) currently supported. Rule-based agents provide lightweight deterministic baselines based on fixed heuristics.

Despite these different backends, all agents interact through a unified action interface. LLM observations are rendered as text and their responses are parsed into continuous action vectors, while RL agents operate directly on numerical observations and actions. Both are mapped to the same action space, allowing the same entity logic to process decisions regardless of backend. This design keeps table schemas and referential structure fixed, enables controlled compar-

isons across agent types, and allows LLM and RL agents to coexist within the same simulation.

### 2.4. Persona Layer

To increase behavioral diversity, TableFactory includes a persona layer based on the Myers–Briggs Type Indicator (MBTI) (Myers, 2003; Schweiger, 1985), a personality framework that groups individuals into 16 types defined by four dichotomies: introversion/extraversion, sensing/intuition, thinking/feeling, and judging/perceiving. In our system, a sampled MBTI type is mapped to latent behavioral traits such as patience, price sensitivity, exploration tendency, empathy, politeness, and complaint frequency. These traits are then injected into agents through backend-specific channels. For example, in the butcher-shop scenario, a customer with high patience and a constructive complaint style may tolerate a long queue but provide detailed feedback if product quality is poor. For LLM agents, these traits are expressed through persona descriptions in the prompt; for RL agents, they are appended as numerical inputs to the observation space. This design allows personas to shape behavior across agent families without changing the underlying business logic or table schema.

### 2.5. Runner and Training Loop

The runner executes the simulation over a sequence of business days and handles both data generation and optional policy training. In each day, entities receive observations from the current world state, agents act, and the resulting business events update the state and write rows to the output tables. For example, in the butcher-shop scenario, higher

*Table 2.* Simulation results on the Barcelona butcher-shop scenario (7 days). Higher is better (↑) except Support Tickets (↓).

| Metric | GPT-5.4 | PPO | Rule-based |
|---|---|---|---|
| Total Orders | 327 | 38 | 307 |
| Total Revenue | €17,045 | €1,463 | €13,900 |
| Avg. Order Value | €52.13 | €38.51 | €45.28 |
| Purchase Entropy | 4.92 | 4.49 | 0.00 |
| Support Tickets | 11 | 236 | 15 |

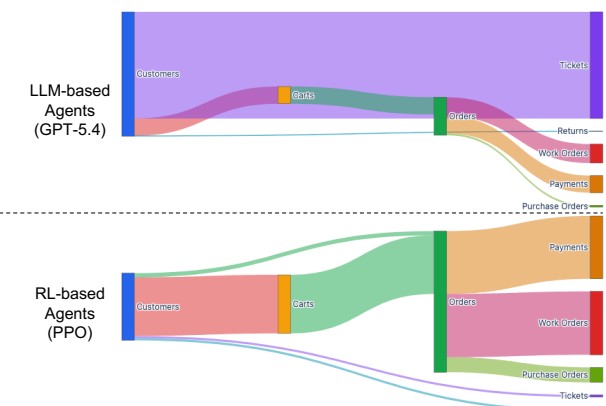

*Figure 2.* **Customer decision flow in the Berlin computer store scenario over 7 days:** LLM (top) vs. RL (bottom).

weekend demand increases customer arrivals, leading Sales to process more orders, Operations to fill preparation work orders, Inventory to trigger replenishment, and Manager to handle escalations. For RL agents, the runner also stores transitions and updates policies during training. Regardless of backend, the exported artifacts follow the same format: linked tables covering master data, commerce, inventory and procurement, customer support, and operations, together with configuration snapshots, run metadata, metrics histories, and persona profiles when enabled.

### 2.6. User Customization

TableFactory supports three levels of control: fully automatic generation from a natural-language prompt, selective editing of generated fields (*e.g.*, products, prices, demand parameters), or a complete user-supplied YAML configuration. All three modes share the same execution pipeline.

## 3. Results

**Qualitative Results.** Due to space constraints, we present the full generated tables in Sec. A. The appendix showcases seven diverse Berlin business scenarios (butcher shop, beer garden, cafe, computer store, fruit & vegetable shop, kebab food truck, vintage clothing shop), each with eight foreign-key-linked output tables whose referential integrity can be traced across rows, e.g., a `customer_id` in an order appears in the corresponding payment and shipment.

*Table 3.* Comparison of LLM backends on the Barcelona butcher-shop scenario (7 days).

| Model | Orders ↑ | Revenue ↑ | Products ↑ | Categories ↑ | Avg. Order Value ↑ |
|---|---|---|---|---|---|
| gpt-4o | 385 | €43,075 | 10 | 7 | €111.88 |
| gpt-5-mini | 439 | €18,719 | 20 | 12 | €42.64 |
| gpt-5.4 | 350 | €15,261 | 31 | 12 | €43.60 |
| claude-sonnet-4-6 | 377 | €12,140 | 74 | 9 | €32.20 |
| claude-opus-4-6 | 420 | €13,780 | 50 | 8 | €32.81 |
| gemini-3-flash | 309 | €16,403 | 18 | 6 | €53.08 |

*Table 4.* Seven Berlin business types simulated with GPT-5.4 (7 days each, same codebase, zero code changes).

| Shop | Products | Avg. Price | Orders ↑ | Revenue ↑ | Avg. Order Value ↑ |
|---|---|---|---|---|---|
| Beer Garden | 28 | €5.91 | 497 | €8,785 | €17.68 |
| Butcher | 31 | €8.02 | 311 | €7,661 | €24.63 |
| Cafe | 30 | €4.60 | 444 | €6,135 | €13.82 |
| Computer Store | 38 | €436.79 | 250 | €300,247 | €1,200.99 |
| Veg & Fruits | 36 | €4.49 | 315 | €4,672 | €14.83 |
| Kebab Truck | 28 | €0.80–9.50 | 433 | €5,890 | €13.60 |
| Vintage Clothes | 24 | €62.75 | 229 | €40,363 | €176.26 |

**Study of agent interchangeability.** Tab. 2 compares LLM (GPT-5.4), RL (PPO), and rule-based agents on the same Barcelona butcher-shop scenario; all three produce valid simulations with identical output schemas. GPT-5.4 achieves the highest order volume (327), revenue (€17,045), and purchase entropy (4.92). The rule-based agent matches on order volume (307) but with zero entropy, reflecting deterministic product selection. PPO, with only 7 days of online training, generates 38 orders, consistent with the short training horizon. Fig. 2 visualizes the distinct customer decision paths produced by LLM vs. RL agents in a Berlin computer store simulation.

**Cross-model comparison.** Tab. 3 compares six LLM backends on the same Barcelona butcher-shop. All produce valid simulations with distinct trade-offs: gpt-4o achieves the highest per-order revenue (€111.88) but only 10 products, while claude-sonnet-4-6 covers 74 products at lower order value. Users can swap backends without code changes.

**Diverse business scenarios from free-form prompts.** Tab. 4 reports seven Berlin business types, each from a single prompt with zero code changes. Generated catalogs reflect realistic pricing: a cafe averages €4.60 per product vs. €436.79 for a computer store. Revenue scales accordingly, high-frequency low-ticket businesses (beer garden: 497 orders, €8,785) contrast with low-frequency high-ticket ones (computer store: 250 orders, €300,247).

## 4. Conclusion

We presented **TableFactory**, a framework for generating semantically linked relational tables through multi-agent behavioral simulation over a shared world state. It unifies prompt-conditioned scenarios, abstract roles, LLM or RL controllers, and personas in a single pipeline. Our diagnostic results show controller interchangeability, scenario and regional variation, and persona-driven diversity. TableFactory provides a practical foundation for generating linked synthetic data with cross-table consistency.

## Impact Statement

This paper presents **TableFactory**, a framework for generating semantically linked multi-table tabular data through multi-agent behavioral simulation. Its main positive impact is to support research on tabular learning, relational reasoning, and enterprise-style data systems in settings where real operational data is difficult to access because of privacy, confidentiality, and limited sharing. At the same time, systems for synthetic business data generation may be misused to create misleading records, unrealistic benchmarks, or data that reflects biases introduced by prompts, personas, external grounding, or agent policies. Our framework is intended as a research tool for simulation and data generation, not as a substitute for audited real-world data in high-stakes decision making. Responsible use should therefore include clear disclosure that the data is synthetic, careful validation before downstream deployment, and attention to fairness, privacy, and misuse risks when extending or applying the system.

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

## Appendix Overview

This appendix provides comprehensive supplementary material for TableFactory. App. A presents qualitative results: raw output tables from seven diverse Berlin-based business scenarios, allowing the reader to inspect generated data and trace foreign-key linkages across tables. The remaining sections detail the system's implementation. App. B describes the scenario generation pipeline, including location context, demand modeling, and the LLM-based reward model. App. C and App. D cover the LLM-based and RL-based agent implementations, respectively, including prompt construction, response parsing, network architectures, and hyperparameters. App. E explains the MBTI persona layer that introduces behavioral diversity across agents. App. F formalizes the observation and action spaces, the MDP formulation, and per-entity reward functions. App. G and App. H document the entity layer design and inter-entity interaction pathways mediated through the shared WorldState. Finally, App. I describes the output schema and causal data chain, and App. J summarizes the codebase organization.

## A. Qualitative Results

This section presents raw output tables generated by TableFactory for seven diverse Berlin-based business scenarios, all produced with the same codebase using GPT-5.4 as the LLM agent backend over seven simulated days (seed 42). For each scenario, we show five representative rows from eight core tables: *products*, *suppliers*, *customers*, *orders*, *order_items*, *payments*, *shipments*, and *work_orders*. The reader can trace foreign-key linkages across tables, for example, a customer_id appearing in an order row also appears in the corresponding payment and shipment rows, and a product_id in the order-items table references a row in the products table. For brevity, created_at and updated_at timestamp columns are omitted from all tables, and all timestamps are shortened to date-only format.

### A.1. Butcher Shop: Kiez Fleischerei Prenzlauer

The input prompt for this scenario was: *"simulate a butcher shop in Berlin, Germany"*. TableFactory generated the store name *Kiez Fleischerei Prenzlauer* along with a complete business configuration. The simulation used an LLM agent (GPT-5.4) over 7 days, producing 31 products, 332 customers, and 311 orders. Tables Tab. 5–Tab. 12 show representative rows from the eight core output tables.

*Table 5.* Products table for the Berlin Butcher Shop scenario (5 of 31 rows shown).

| ID | Name | Category | Price | Cost | Supplier | WO | Type |
|----|------|----------|-------|------|----------|-----|------|
| BEEF01 | Rinderhackfleisch 500g | beef | 7.49 | 4.35 | SUP1 | Yes | mincing |
| PORK05 | Kasseler Kotelett 500g | pork | 7.50 | 4.45 | SUP2 | No | – |
| LAMB02 | Lammhack 500g | lamb | 9.90 | 6.10 | SUP4 | Yes | mincing |
| DELI01 | Frikadellen 2 Stk | deli_prepared | 4.50 | 2.35 | SUP1 | Yes | cooking |
| SPEC03 | Rinderknochen für Brühe 1kg | specialty | 4.50 | 2.20 | SUP1 | No | – |

*Table 6.* Suppliers table for the Berlin Butcher Shop scenario (all 4 rows).

| ID | Name | Lead Time (days) | MOQ | Reliability |
|----|------|------------------|-----|-------------|
| SUP1 | Brandenburger Rind & Kalb GmbH | 2 | 10 | 0.95 |
| SUP2 | Fleischgroßmarkt Berlin Süd | 1 | 12 | 0.93 |
| SUP3 | Geflügelhof Havelland KG | 2 | 8 | 0.91 |
| SUP4 | Märkische Weidelamm OHG | 3 | 6 | 0.88 |

*Table 7.* Customers table for the Berlin Butcher Shop scenario (5 of 332 rows shown).

| ID | Segment | Address |
|----|---------|---------|
| CUS-000001 | consumer | 88 Main St, Berlin |
| CUS-000084 | consumer | 76 Main St, Berlin |
| CUS-000167 | consumer | 184 Main St, Berlin |
| CUS-000249 | consumer | 16 Main St, Berlin |
| CUS-000332 | consumer | 22 Main St, Berlin |

*Table 8.* Orders table for the Berlin Butcher Shop scenario (5 of 311 rows shown).

| Order ID | Customer ID | Date | Status | Payment |
|---|---|---|---|---|
| ORD-000001 | CUS-000001 | 2025-01-01 | shipped | credit_card |
| ORD-000079 | CUS-000079 | 2025-01-03 | shipped | credit_card |
| ORD-000156 | CUS-000152 | 2025-01-04 | shipped | credit_card |
| ORD-000233 | CUS-000224 | 2025-01-05 | shipped | credit_card |
| ORD-000311 | CUS-000079 | 2025-01-07 | paid | credit_card |

*Table 9.* Order Items table for the Berlin Butcher Shop scenario (5 of 630 rows shown).

| Order ID | Product ID | Qty | Unit Price |
|---|---|---|---|
| ORD-000001 | COLD03 | 1 | 4.90 |
| ORD-000076 | COLD03 | 2 | 4.90 |
| ORD-000161 | PORK04 | 2 | 6.50 |
| ORD-000240 | BEEF02 | 1 | 12.90 |
| ORD-000311 | BEEF01 | 2 | 7.49 |

*Table 10.* Payments table for the Berlin Butcher Shop scenario (5 of 311 rows shown).

| Payment ID | Order ID | Amount | Method | Status | Date |
|---|---|---|---|---|---|
| PAY-000001 | ORD-000001 | 4.90 | credit_card | captured | 2025-01-01 |
| PAY-000079 | ORD-000079 | 39.68 | credit_card | captured | 2025-01-03 |
| PAY-000156 | ORD-000156 | 11.30 | credit_card | captured | 2025-01-04 |
| PAY-000233 | ORD-000233 | 23.50 | credit_card | captured | 2025-01-05 |
| PAY-000311 | ORD-000311 | 39.68 | credit_card | captured | 2025-01-07 |

*Table 11.* Shipments table for the Berlin Butcher Shop scenario (5 of 306 rows shown).

| Shipment ID | Order ID | Carrier | Tracking No. | Shipped | Est. Delivery | Status |
|---|---|---|---|---|---|---|
| SHP-000001 | ORD-000001 | DPD | TRK3823637 | 2025-01-01 | 2025-01-03 | shipped |
| SHP-000077 | ORD-000077 | GLS | TRK6128777 | 2025-01-03 | 2025-01-05 | shipped |
| SHP-000153 | ORD-000153 | DHL | TRK4169912 | 2025-01-04 | 2025-01-06 | shipped |
| SHP-000230 | ORD-000230 | DPD | TRK7804676 | 2025-01-05 | 2025-01-08 | shipped |
| SHP-000306 | ORD-000306 | UPS | TRK1212097 | 2025-01-07 | 2025-01-10 | shipped |

*Table 12.* Work Orders table for the Berlin Butcher Shop scenario (5 of 360 rows shown).

| WO ID | Customer ID | Date | Type | Status |
|---|---|---|---|---|
| WO-000001 | CUS-000002 | 2025-01-01 | forming | completed |
| WO-000091 | CUS-000066 | 2025-01-02 | mincing | new |
| WO-000181 | CUS-000144 | 2025-01-04 | cutting | new |
| WO-000270 | CUS-000224 | 2025-01-05 | mincing | new |
| WO-000360 | CUS-000287 | 2025-01-07 | cutting | new |

## A.2. Beer Garden: Spreegarten am Holzmarkt

The input prompt for this scenario was: *"simulate a beer garden in Berlin, Germany"*. TableFactory generated the store name *Spreegarten am Holzmarkt* along with a complete business configuration. The simulation used an LLM agent (GPT-5.4) over 7 days, producing 28 products, 521 customers, and 497 orders. Tables Tab. 13–Tab. 20 show representative rows from the eight core output tables.

## A.3. Cafe: Kreuzberg Kaffeestube

The input prompt for this scenario was: *"simulate a cafe in Berlin, Germany"*. TableFactory generated the store name *Kreuzberg Kaffeestube* along with a complete business configuration. The simulation used an LLM agent (GPT-5.4) over 7 days, producing 30 products, 479 customers, and 444 orders. Tables Tab. 21–Tab. 28 show representative rows from the eight core output tables.

*Table 13.* Products table for the Berlin Beer Garden scenario (5 of 28 rows shown).

| ID | Name | Category | Price | Cost | Supplier | WO | Type |
|----|------|----------|-------|------|----------|----|------|
| BEER01 | Helles Lager 0.5L | draft_beer | 5.20 | 1.35 | SUP01 | Yes | pouring |
| BEER08 | Berliner Weisse 0.33L | craft_beer | 5.90 | 2.10 | SUP02 | No | – |
| NA03 | Sparkling Water 0.33L | soft_drink | 3.40 | 0.65 | SUP04 | No | – |
| FOOD05 | Potato Salad Bowl | cold_food | 5.90 | 1.90 | SUP02 | Yes | assembly |
| DESS02 | Kaiserschmarrn Cup | dessert | 6.40 | 2.10 | SUP02 | Yes | cooking |

*Table 14.* Suppliers table for the Berlin Beer Garden scenario (all 4 rows).

| ID | Name | Lead Time (days) | MOQ | Reliability |
|----|------|------------------|-----|-------------|
| SUP01 | Berliner Brauhaus Vertrieb GmbH | 2 | 20 | 0.95 |
| SUP02 | Markthalle Neun Gastroservice | 2 | 15 | 0.91 |
| SUP03 | Wein & Spirituosen Kontor Berlin | 3 | 12 | 0.89 |
| SUP04 | Spree Getränke Logistik | 1 | 24 | 0.94 |

*Table 15.* Customers table for the Berlin Beer Garden scenario (5 of 521 rows shown).

| ID | Segment | Address |
|----|---------|---------|
| CUS-000001 | consumer | 88 Main St, Berlin |
| CUS-000131 | consumer | 115 Main St, Berlin |
| CUS-000261 | consumer | 184 Main St, Berlin |
| CUS-000391 | consumer | 192 Main St, Berlin |
| CUS-000521 | consumer | 81 Main St, Berlin |

*Table 16.* Orders table for the Berlin Beer Garden scenario (5 of 497 rows shown).

| Order ID | Customer ID | Date | Status | Payment |
|----------|-------------|------|--------|---------|
| ORD-000001 | CUS-000001 | 2025-01-01 | shipped | credit_card |
| ORD-000125 | CUS-000124 | 2025-01-03 | shipped | credit_card |
| ORD-000249 | CUS-000246 | 2025-01-04 | shipped | credit_card |
| ORD-000373 | CUS-000365 | 2025-01-05 | shipped | credit_card |
| ORD-000497 | CUS-000156 | 2025-01-07 | paid | credit_card |

*Table 17.* Order Items table for the Berlin Beer Garden scenario (5 of 993 rows shown).

| Order ID | Product ID | Qty | Unit Price |
|----------|------------|-----|------------|
| ORD-000001 | FOOD04 | 1 | 7.40 |
| ORD-000128 | FOOD01 | 2 | 6.90 |
| ORD-000253 | BEER04 | 2 | 5.80 |
| ORD-000377 | NA04 | 2 | 4.10 |
| ORD-000497 | FOOD07 | 2 | 7.20 |

*Table 18.* Payments table for the Berlin Beer Garden scenario (5 of 497 rows shown).

| Payment ID | Order ID | Amount | Method | Status | Date |
|------------|----------|--------|--------|--------|------|
| PAY-000001 | ORD-000001 | 7.40 | credit_card | captured | 2025-01-01 |
| PAY-000125 | ORD-000125 | 5.60 | credit_card | captured | 2025-01-03 |
| PAY-000249 | ORD-000249 | 10.60 | credit_card | captured | 2025-01-04 |
| PAY-000373 | ORD-000373 | 20.00 | credit_card | captured | 2025-01-05 |
| PAY-000497 | ORD-000497 | 30.40 | credit_card | captured | 2025-01-07 |

### A.4. Computer Store: SpreeTech Berlin

The input prompt for this scenario was: *"simulate a computer store in Berlin, Germany"*. TableFactory generated the store name *SpreeTech Berlin* along with a complete business configuration. The simulation used an LLM agent (GPT-5.4) over 7 days, producing 38 products, 250 customers, and 250 orders. Tables Tab. 29–Tab. 36 show representative rows from the

*Table 19.* Shipments table for the Berlin Beer Garden scenario (5 of 492 rows shown).

| Shipment ID | Order ID | Carrier | Tracking No. | Shipped | Est. Delivery | Status |
|---|---|---|---|---|---|---|
| SHP-000001 | ORD-000001 | DHL | TRK6437597 | 2025-01-01 | 2025-01-03 | shipped |
| SHP-000124 | ORD-000124 | DPD | TRK6746207 | 2025-01-03 | 2025-01-04 | shipped |
| SHP-000247 | ORD-000247 | DHL | TRK2748375 | 2025-01-04 | 2025-01-05 | shipped |
| SHP-000369 | ORD-000369 | DHL | TRK1620450 | 2025-01-05 | 2025-01-06 | shipped |
| SHP-000492 | ORD-000492 | Hermes | TRK4750308 | 2025-01-07 | 2025-01-08 | shipped |

*Table 20.* Work Orders table for the Berlin Beer Garden scenario (5 of 672 rows shown).

| WO ID | Customer ID | Date | Type | Status |
|---|---|---|---|---|
| WO-000001 | CUS-000001 | 2025-01-01 | assembly | completed |
| WO-000169 | CUS-000122 | 2025-01-03 | cooking | new |
| WO-000337 | CUS-000245 | 2025-01-04 | pouring | new |
| WO-000504 | CUS-000360 | 2025-01-05 | cooking | new |
| WO-000672 | CUS-000474 | 2025-01-07 | mixing | new |

*Table 21.* Products table for the Berlin Cafe scenario (5 of 30 rows shown).

| ID | Name | Category | Price | Cost | Supplier | WO | Type |
|---|---|---|---|---|---|---|---|
| ESP | Espresso | hot_coffee | 2.40 | 0.45 | SUP1 | Yes | coffee_preparation |
| CHAI | Chai Latte | hot_drinks | 4.50 | 1.20 | SUP3 | Yes | drink_preparation |
| OJ | Fresh Orange Juice | cold_drinks | 4.90 | 1.80 | SUP4 | Yes | drink_preparation |
| COOKIE | Chocolate Chip Cookie | snacks | 2.40 | 0.85 | SUP2 | No | – |
| QUICHE | Spinach Feta Quiche Slice | savory_bakes | 5.80 | 2.30 | SUP2 | No | – |

*Table 22.* Suppliers table for the Berlin Cafe scenario (all 4 rows).

| ID | Name | Lead Time (days) | MOQ | Reliability |
|---|---|---|---|---|
| SUP1 | Berliner Bohnen Rösterei GmbH | 3 | 10 | 0.94 |
| SUP2 | Spree Backwaren Lieferdienst | 2 | 15 | 0.91 |
| SUP3 | Mitte Tee & Spezialitäten Handel | 4 | 8 | 0.89 |
| SUP4 | Markthalle Frischelogistik Berlin | 1 | 12 | 0.93 |

*Table 23.* Customers table for the Berlin Cafe scenario (5 of 479 rows shown).

| ID | Segment | Address |
|---|---|---|
| CUS-000001 | consumer | 88 Main St, Berlin |
| CUS-000121 | consumer | 160 Main St, Berlin |
| CUS-000240 | consumer | 53 Main St, Berlin |
| CUS-000359 | b2b | 4 Main St, Berlin |
| CUS-000479 | consumer | 174 Main St, Berlin |

*Table 24.* Orders table for the Berlin Cafe scenario (5 of 444 rows shown).

| Order ID | Customer ID | Date | Status | Payment |
|---|---|---|---|---|
| ORD-000001 | CUS-000001 | 2025-01-01 | shipped | credit_card |
| ORD-000112 | CUS-000112 | 2025-01-02 | shipped | credit_card |
| ORD-000223 | CUS-000217 | 2025-01-04 | shipped | credit_card |
| ORD-000333 | CUS-000317 | 2025-01-06 | shipped | credit_card |
| ORD-000444 | CUS-000169 | 2025-01-07 | paid | credit_card |

eight core output tables.

### A.5. Fruit & Vegetable Shop: Spree Frucht & Gemüse

The input prompt for this scenario was: *"simulate a vegetable and fruits shop in Berlin, Germany"*. TableFactory generated the store name *Spree Frucht & Gemüse* along with a complete business configuration. The simulation used an LLM agent

*Table 25.* Order Items table for the Berlin Cafe scenario (5 of 899 rows shown).

| Order ID | Product ID | Qty | Unit Price |
|---|---|---|---|
| ORD-000001 | CHEESE | 1 | 4.90 |
| ORD-000111 | FLT | 2 | 4.30 |
| ORD-000222 | ICELAT | 1 | 4.60 |
| ORD-000335 | AVOTOAST | 1 | 8.90 |
| ORD-000444 | AVOTOAST | 1 | 8.90 |

*Table 26.* Payments table for the Berlin Cafe scenario (5 of 444 rows shown).

| Payment ID | Order ID | Amount | Method | Status | Date |
|---|---|---|---|---|---|
| PAY-000001 | ORD-000001 | 4.90 | credit_card | captured | 2025-01-01 |
| PAY-000112 | ORD-000112 | 16.40 | credit_card | captured | 2025-01-02 |
| PAY-000223 | ORD-000223 | 11.40 | credit_card | captured | 2025-01-04 |
| PAY-000333 | ORD-000333 | 11.40 | credit_card | captured | 2025-01-06 |
| PAY-000444 | ORD-000444 | 20.90 | credit_card | captured | 2025-01-07 |

*Table 27.* Shipments table for the Berlin Cafe scenario (5 of 439 rows shown).

| Shipment ID | Order ID | Carrier | Tracking No. | Shipped | Est. Delivery | Status |
|---|---|---|---|---|---|---|
| SHP-000001 | ORD-000001 | Hermes | TRK9382341 | 2025-01-01 | 2025-01-02 | shipped |
| SHP-000111 | ORD-000111 | DPD | TRK5054146 | 2025-01-02 | 2025-01-03 | shipped |
| SHP-000220 | ORD-000220 | DHL | TRK4161005 | 2025-01-04 | 2025-01-06 | shipped |
| SHP-000329 | ORD-000329 | DPD | TRK4778294 | 2025-01-06 | 2025-01-07 | shipped |
| SHP-000439 | ORD-000439 | Hermes | TRK1274206 | 2025-01-07 | 2025-01-10 | shipped |

*Table 28.* Work Orders table for the Berlin Cafe scenario (5 of 625 rows shown).

| WO ID | Customer ID | Date | Type | Status |
|---|---|---|---|---|
| WO-000001 | CUS-000002 | 2025-01-01 | assembly | completed |
| WO-000157 | CUS-000113 | 2025-01-02 | drink_preparation | new |
| WO-000313 | CUS-000213 | 2025-01-04 | coffee_preparation | new |
| WO-000469 | CUS-000317 | 2025-01-06 | assembly | new |
| WO-000625 | CUS-000418 | 2025-01-07 | coffee_preparation | new |

*Table 29.* Products table for the Berlin Computer Store scenario (5 of 38 rows shown).

| ID | Name | Category | Price | Cost | Supplier | WO | Type |
|---|---|---|---|---|---|---|---|
| LAP001 | Lenovo IdeaPad 15 Gen 8 | laptops | 699.00 | 548.00 | SUP1 | No | – |
| MON003 | ASUS 32 inch 165Hz Gaming Monitor | monitors | 399.00 | 302.00 | SUP2 | No | – |
| RAM001 | 16GB DDR4 RAM Kit | components | 49.00 | 33.00 | SUP3 | No | – |
| ACC002 | Laptop Backpack 15.6 inch | accessories | 39.00 | 22.00 | SUP4 | No | – |
| BND002 | Student Laptop Setup Bundle | bundles | 799.00 | 603.00 | SUP1 | Yes | assembly |

*Table 30.* Suppliers table for the Berlin Computer Store scenario (all 4 rows).

| ID | Name | Lead Time (days) | MOQ | Reliability |
|---|---|---|---|---|
| SUP1 | TechData Deutschland GmbH | 3 | 5 | 0.95 |
| SUP2 | Ingram Micro Distribution Berlin | 4 | 4 | 0.93 |
| SUP3 | ByteWerk Komponentenhandel | 2 | 3 | 0.91 |
| SUP4 | Kreuzberg Peripherals Supply | 2 | 10 | 0.89 |

(GPT-5.4) over 7 days, producing 36 products, 343 customers, and 315 orders. Tables Tab. 37–Tab. 44 show representative rows from the eight core output tables.

*Table 31.* Customers table for the Berlin Computer Store scenario (5 of 250 rows shown).

| ID | Segment | Address |
|---|---|---|
| CUS-000001 | consumer | 88 Main St, Berlin |
| CUS-000063 | consumer | 6 Main St, Berlin |
| CUS-000125 | consumer | 117 Main St, Berlin |
| CUS-000188 | consumer | 194 Main St, Berlin |
| CUS-000250 | consumer | 159 Main St, Berlin |

*Table 32.* Orders table for the Berlin Computer Store scenario (5 of 250 rows shown).

| Order ID | Customer ID | Date | Status | Payment |
|---|---|---|---|---|
| ORD-000001 | CUS-000001 | 2025-01-01 | shipped | credit_card |
| ORD-000063 | CUS-000016 | 2025-01-02 | shipped | credit_card |
| ORD-000125 | CUS-000115 | 2025-01-04 | shipped | credit_card |
| ORD-000188 | CUS-000168 | 2025-01-06 | shipped | credit_card |
| ORD-000250 | CUS-000027 | 2025-01-07 | paid | credit_card |

*Table 33.* Order Items table for the Berlin Computer Store scenario (5 of 501 rows shown).

| Order ID | Product ID | Qty | Unit Price |
|---|---|---|---|
| ORD-000001 | PRN002 | 1 | 99.00 |
| ORD-000066 | ACC002 | 2 | 39.00 |
| ORD-000127 | KEY002 | 2 | 89.99 |
| ORD-000187 | ACC002 | 2 | 39.00 |
| ORD-000250 | MON001 | 2 | 149.00 |

*Table 34.* Payments table for the Berlin Computer Store scenario (5 of 250 rows shown).

| Payment ID | Order ID | Amount | Method | Status | Date |
|---|---|---|---|---|---|
| PAY-000001 | ORD-000001 | 99.00 | credit_card | captured | 2025-01-01 |
| PAY-000063 | ORD-000063 | 208.98 | credit_card | captured | 2025-01-02 |
| PAY-000125 | ORD-000125 | 1599.00 | credit_card | captured | 2025-01-04 |
| PAY-000188 | ORD-000188 | 159.98 | credit_card | captured | 2025-01-06 |
| PAY-000250 | ORD-000250 | 626.98 | credit_card | captured | 2025-01-07 |

*Table 35.* Shipments table for the Berlin Computer Store scenario (5 of 245 rows shown).

| Shipment ID | Order ID | Carrier | Tracking No. | Shipped | Est. Delivery | Status |
|---|---|---|---|---|---|---|
| SHP-000001 | ORD-000001 | DHL | TRK2481470 | 2025-01-01 | 2025-01-02 | shipped |
| SHP-000062 | ORD-000062 | Hermes | TRK5538515 | 2025-01-03 | 2025-01-04 | shipped |
| SHP-000123 | ORD-000123 | GLS | TRK2852377 | 2025-01-04 | 2025-01-06 | shipped |
| SHP-000184 | ORD-000184 | GLS | TRK7205279 | 2025-01-06 | 2025-01-08 | shipped |
| SHP-000245 | ORD-000245 | Hermes | TRK4638487 | 2025-01-07 | 2025-01-08 | shipped |

*Table 36.* Work Orders table for the Berlin Computer Store scenario (5 of 111 rows shown).

| WO ID | Customer ID | Date | Type | Status |
|---|---|---|---|---|
| WO-000001 | CUS-000002 | 2025-01-01 | assembly | completed |
| WO-000029 | CUS-000063 | 2025-01-03 | assembly | completed |
| WO-000056 | CUS-000109 | 2025-01-04 | assembly | new |
| WO-000083 | CUS-000164 | 2025-01-05 | assembly | new |
| WO-000111 | CUS-000215 | 2025-01-07 | assembly | new |

### A.6. Kebab Food Truck: Spree Grill Döner

The input prompt for this scenario was: *"simulate a kebab food truck in Berlin, Germany"*. TableFactory generated the store name *Spree Grill Döner* along with a complete business configuration. The simulation used an LLM agent (GPT-5.4) over

*Table 37.* Products table for the Berlin Fruit & Vegetable Shop scenario (5 of 36 rows shown).

| ID | Name | Category | Price | Cost | Supplier | WO | Type |
|----|------|----------|-------|------|----------|----|----|
| APL_GALA | Gala Apples 1 kg | fresh_fruit | 2.99 | 1.65 | SUP_REGIO | No | – |
| MAN_1PC | Mango each | tropical_fruit | 2.49 | 1.42 | SUP_TROPIC | No | – |
| BRC_500 | Broccoli 500 g | vegetables | 2.19 | 1.12 | SUP_REGIO | No | – |
| SAL_LARGE | Large Salad Mix Bowl | prepared_fresh | 7.49 | 3.18 | SUP_REGIO | Yes | assembly |
| DRY_DATES | Medjool Dates 250 g | dried_fruit | 5.49 | 3.12 | SUP_TROPIC | No | – |

*Table 38.* Suppliers table for the Berlin Fruit & Vegetable Shop scenario (all 3 rows).

| ID | Name | Lead Time (days) | MOQ | Reliability |
|----|------|------------------|-----|-------------|
| SUP_REGIO | BrandenFrisch Großmarkt GmbH | 1 | 20 | 0.94 |
| SUP_TROPIC | Berlin Tropenimport Handels KG | 2 | 15 | 0.9 |
| SUP_HERB | Kräuterhof Potsdam eG | 1 | 10 | 0.96 |

*Table 39.* Customers table for the Berlin Fruit & Vegetable Shop scenario (5 of 343 rows shown).

| ID | Segment | Address |
|----|---------|---------|
| CUS-000001 | consumer | 88 Main St, Berlin |
| CUS-000087 | consumer | 176 Main St, Berlin |
| CUS-000172 | consumer | 39 Main St, Berlin |
| CUS-000257 | consumer | 7 Main St, Berlin |
| CUS-000343 | consumer | 42 Main St, Berlin |

*Table 40.* Orders table for the Berlin Fruit & Vegetable Shop scenario (5 of 315 rows shown).

| Order ID | Customer ID | Date | Status | Payment |
|----------|-------------|------|--------|---------|
| ORD-000001 | CUS-000001 | 2025-01-01 | shipped | credit_card |
| ORD-000079 | CUS-000079 | 2025-01-02 | shipped | credit_card |
| ORD-000158 | CUS-000153 | 2025-01-04 | shipped | credit_card |
| ORD-000237 | CUS-000223 | 2025-01-06 | shipped | credit_card |
| ORD-000315 | CUS-000056 | 2025-01-07 | paid | credit_card |

*Table 41.* Order Items table for the Berlin Fruit & Vegetable Shop scenario (5 of 631 rows shown).

| Order ID | Product ID | Qty | Unit Price |
|----------|------------|-----|------------|
| ORD-000001 | SAL_SMALL | 1 | 4.99 |
| ORD-000078 | ORG_NAVL | 1 | 3.49 |
| ORD-000163 | AVO_HASS | 1 | 3.49 |
| ORD-000239 | LET_ROM | 1 | 1.79 |
| ORD-000315 | FRUIT_BOX | 2 | 24.90 |

*Table 42.* Payments table for the Berlin Fruit & Vegetable Shop scenario (5 of 315 rows shown).

| Payment ID | Order ID | Amount | Method | Status | Date |
|------------|----------|--------|--------|--------|------|
| PAY-000001 | ORD-000001 | 4.99 | credit_card | captured | 2025-01-01 |
| PAY-000079 | ORD-000079 | 11.97 | credit_card | captured | 2025-01-02 |
| PAY-000158 | ORD-000158 | 6.98 | credit_card | captured | 2025-01-04 |
| PAY-000237 | ORD-000237 | 8.06 | credit_card | captured | 2025-01-06 |
| PAY-000315 | ORD-000315 | 54.27 | credit_card | captured | 2025-01-07 |

7 days, producing 28 products, 474 customers, and 433 orders. Tables Tab. 45–Tab. 52 show representative rows from the eight core output tables.

*Table 43.* Shipments table for the Berlin Fruit & Vegetable Shop scenario (5 of 312 rows shown).

| Shipment ID | Order ID | Carrier | Tracking No. | Shipped | Est. Delivery | Status |
|---|---|---|---|---|---|---|
| SHP-000001 | ORD-000001 | DHL | TRK7381324 | 2025-01-01 | 2025-01-02 | shipped |
| SHP-000079 | ORD-000079 | DHL | TRK5525924 | 2025-01-02 | 2025-01-03 | shipped |
| SHP-000157 | ORD-000157 | DPD | TRK6181303 | 2025-01-04 | 2025-01-06 | shipped |
| SHP-000234 | ORD-000234 | DHL | TRK4621337 | 2025-01-06 | 2025-01-09 | shipped |
| SHP-000312 | ORD-000312 | UPS | TRK1039272 | 2025-01-07 | 2025-01-10 | shipped |

*Table 44.* Work Orders table for the Berlin Fruit & Vegetable Shop scenario (5 of 176 rows shown).

| WO ID | Customer ID | Date | Type | Status |
|---|---|---|---|---|
| WO-000001 | CUS-000001 | 2025-01-01 | assembly | completed |
| WO-000045 | CUS-000067 | 2025-01-02 | assembly | new |
| WO-000089 | CUS-000160 | 2025-01-04 | cooking | new |
| WO-000132 | CUS-000217 | 2025-01-06 | assembly | new |
| WO-000176 | CUS-000293 | 2025-01-07 | assembly | new |

*Table 45.* Products table for the Berlin Kebab Food Truck scenario (5 of 28 rows shown).

| ID | Name | Category | Price | Cost | Supplier | WO | Type |
|---|---|---|---|---|---|---|---|
| P001 | Classic Chicken Döner | döner | 7.50 | 2.60 | SUP1 | Yes | cooking |
| P008 | Chicken Döner Box | box_meal | 8.50 | 3.00 | SUP1 | Yes | assembly |
| P015 | Extra Meat | add_on | 2.50 | 1.00 | SUP1 | Yes | assembly |
| P021 | Cola Zero 330ml | drink | 2.80 | 1.00 | SUP4 | No | – |
| P028 | Baklava 2 pcs | dessert | 3.50 | 1.20 | SUP3 | No | – |

*Table 46.* Suppliers table for the Berlin Kebab Food Truck scenario (all 4 rows).

| ID | Name | Lead Time (days) | MOQ | Reliability |
|---|---|---|---|---|
| SUP1 | Berlin Döner Großhandel GmbH | 2 | 20 | 0.93 |
| SUP2 | Metro Gastro Berlin | 2 | 15 | 0.95 |
| SUP3 | Bäckerei & Feinkost Yildiz KG | 1 | 10 | 0.91 |
| SUP4 | Getränke Nordost Berlin | 1 | 24 | 0.96 |

*Table 47.* Customers table for the Berlin Kebab Food Truck scenario (5 of 474 rows shown).

| ID | Segment | Address |
|---|---|---|
| CUS-000001 | consumer | 88 Main St, Berlin |
| CUS-000119 | b2b | 68 Main St, Berlin |
| CUS-000237 | consumer | 16 Main St, Berlin |
| CUS-000356 | consumer | 185 Main St, Berlin |
| CUS-000474 | consumer | 184 Main St, Berlin |

*Table 48.* Orders table for the Berlin Kebab Food Truck scenario (5 of 433 rows shown).

| Order ID | Customer ID | Date | Status | Payment |
|---|---|---|---|---|
| ORD-000001 | CUS-000001 | 2025-01-01 | shipped | credit_card |
| ORD-000109 | CUS-000109 | 2025-01-02 | shipped | credit_card |
| ORD-000217 | CUS-000214 | 2025-01-04 | shipped | credit_card |
| ORD-000325 | CUS-000313 | 2025-01-06 | shipped | credit_card |
| ORD-000433 | CUS-000160 | 2025-01-07 | paid | credit_card |

## A.7. Vintage Clothing Shop: Kreuzberg Kollektiv Vintage

The input prompt for this scenario was: *"simulate a vintage clothes shop in Berlin, Germany"*. TableFactory generated the store name *Kreuzberg Kollektiv Vintage* along with a complete business configuration. The simulation used an LLM agent (GPT-5.4) over 7 days, producing 24 products, 249 customers, and 229 orders. Tables Tab. 53–Tab. 60 show representative rows from the eight core output tables.

*Table 49.* Order Items table for the Berlin Kebab Food Truck scenario (5 of 844 rows shown).

| Order ID | Product ID | Qty | Unit Price |
|----------|-----------|-----|-----------|
| ORD-000001 | P020 | 1 | 2.80 |
| ORD-000107 | P013 | 1 | 5.50 |
| ORD-000217 | P012 | 1 | 6.50 |
| ORD-000326 | P007 | 1 | 8.00 |
| ORD-000433 | P025 | 1 | 2.20 |

*Table 50.* Payments table for the Berlin Kebab Food Truck scenario (5 of 433 rows shown).

| Payment ID | Order ID | Amount | Method | Status | Date |
|-----------|----------|--------|--------|--------|------|
| PAY-000001 | ORD-000001 | 2.80 | credit_card | captured | 2025-01-01 |
| PAY-000109 | ORD-000109 | 5.60 | credit_card | captured | 2025-01-02 |
| PAY-000217 | ORD-000217 | 14.30 | credit_card | captured | 2025-01-04 |
| PAY-000325 | ORD-000325 | 7.00 | credit_card | captured | 2025-01-06 |
| PAY-000433 | ORD-000433 | 19.20 | credit_card | captured | 2025-01-07 |

*Table 51.* Shipments table for the Berlin Kebab Food Truck scenario (5 of 429 rows shown).

| Shipment ID | Order ID | Carrier | Tracking No. | Shipped | Est. Delivery | Status |
|------------|----------|---------|--------------|---------|---------------|--------|
| SHP-000001 | ORD-000001 | Hermes | TRK7549738 | 2025-01-01 | 2025-01-03 | shipped |
| SHP-000108 | ORD-000108 | DPD | TRK9656545 | 2025-01-02 | 2025-01-04 | shipped |
| SHP-000215 | ORD-000215 | DPD | TRK7956055 | 2025-01-04 | 2025-01-05 | shipped |
| SHP-000322 | ORD-000322 | DPD | TRK7051955 | 2025-01-06 | 2025-01-07 | shipped |
| SHP-000429 | ORD-000429 | Hermes | TRK7530640 | 2025-01-07 | 2025-01-09 | shipped |

*Table 52.* Work Orders table for the Berlin Kebab Food Truck scenario (5 of 458 rows shown).

| WO ID | Customer ID | Date | Type | Status |
|-------|-------------|------|------|--------|
| WO-000001 | CUS-000002 | 2025-01-01 | cooking | completed |
| WO-000115 | CUS-000107 | 2025-01-02 | cooking | new |
| WO-000229 | CUS-000212 | 2025-01-04 | assembly | new |
| WO-000344 | CUS-000314 | 2025-01-06 | cooking | new |
| WO-000458 | CUS-000412 | 2025-01-07 | cooking | new |

*Table 53.* Products table for the Berlin Vintage Clothing Shop scenario (5 of 24 rows shown).

| ID | Name | Category | Price | Cost | Supplier | WO | Type |
|----|------|----------|-------|------|----------|-----|------|
| VT001 | 1990s Levi's 501 Jeans | denim | 79.00 | 32.00 | SUP1 | Yes | restoration |
| VT007 | Leather Biker Jacket | jackets | 169.00 | 78.00 | SUP3 | Yes | restoration |
| VT013 | High-Waist Mom Jeans | denim | 64.00 | 24.00 | SUP2 | No | – |
| VT018 | Vintage Hoodie | sweatshirts | 52.00 | 19.00 | SUP4 | No | – |
| VT024 | Vintage Wool Beret | accessories | 27.00 | 9.00 | SUP3 | No | – |

*Table 54.* Suppliers table for the Berlin Vintage Clothing Shop scenario (all 4 rows).

| ID | Name | Lead Time (days) | MOQ | Reliability |
|----|------|------------------|-----|-------------|
| SUP1 | Altstadt Textilhandel Berlin | 5 | 8 | 0.91 |
| SUP2 | Spreeufer Vintage Grosshandel | 7 | 10 | 0.88 |
| SUP3 | Brandenburg Retro Waren | 9 | 6 | 0.85 |
| SUP4 | Neukolln Reuse & Rag Co. | 4 | 12 | 0.93 |

# B. Implementation Details of Scenario Generation

The scenario generator (`scenario_generator.py`) converts a single sentence of user intent into a complete, self-contained simulation configuration through a multi-step LLM pipeline. This section describes each step in detail, supplementing the overview in Sec. 2.

*Table 55.* Customers table for the Berlin Vintage Clothing Shop scenario (5 of 249 rows shown).

| ID | Segment | Address |
|----|---------|---------|
| CUS-000001 | consumer | 88 Main St, Berlin |
| CUS-000063 | consumer | 60 Main St, Berlin |
| CUS-000125 | consumer | 164 Main St, Berlin |
| CUS-000187 | consumer | 89 Main St, Berlin |
| CUS-000249 | consumer | 136 Main St, Berlin |

*Table 56.* Orders table for the Berlin Vintage Clothing Shop scenario (5 of 229 rows shown).

| Order ID | Customer ID | Date | Status | Payment |
|----------|-------------|------|--------|---------|
| ORD-000001 | CUS-000001 | 2025-01-01 | shipped | credit_card |
| ORD-000058 | CUS-000058 | 2025-01-03 | shipped | credit_card |
| ORD-000115 | CUS-000114 | 2025-01-04 | shipped | credit_card |
| ORD-000172 | CUS-000170 | 2025-01-05 | shipped | credit_card |
| ORD-000229 | CUS-000188 | 2025-01-07 | paid | credit_card |

*Table 57.* Order Items table for the Berlin Vintage Clothing Shop scenario (5 of 431 rows shown).

| Order ID | Product ID | Qty | Unit Price |
|----------|-----------|-----|-----------|
| ORD-000001 | VT017 | 1 | 84.00 |
| ORD-000058 | VT005 | 1 | 54.00 |
| ORD-000116 | VT003 | 1 | 119.00 |
| ORD-000169 | VT010 | 2 | 44.00 |
| ORD-000229 | VT022 | 1 | 109.00 |

*Table 58.* Payments table for the Berlin Vintage Clothing Shop scenario (5 of 229 rows shown).

| Payment ID | Order ID | Amount | Method | Status | Date |
|-----------|----------|--------|--------|--------|------|
| PAY-000001 | ORD-000001 | 84.00 | credit_card | captured | 2025-01-01 |
| PAY-000058 | ORD-000058 | 98.00 | credit_card | captured | 2025-01-03 |
| PAY-000115 | ORD-000115 | 266.00 | credit_card | captured | 2025-01-04 |
| PAY-000172 | ORD-000172 | 236.00 | credit_card | captured | 2025-01-05 |
| PAY-000229 | ORD-000229 | 109.00 | credit_card | captured | 2025-01-07 |

*Table 59.* Shipments table for the Berlin Vintage Clothing Shop scenario (5 of 226 rows shown).

| Shipment ID | Order ID | Carrier | Tracking No. | Shipped | Est. Delivery | Status |
|-------------|----------|---------|--------------|---------|---------------|--------|
| SHP-000001 | ORD-000001 | UPS | TRK3886114 | 2025-01-01 | 2025-01-03 | shipped |
| SHP-000057 | ORD-000057 | DHL | TRK1163444 | 2025-01-03 | 2025-01-04 | shipped |
| SHP-000113 | ORD-000113 | UPS | TRK4498372 | 2025-01-04 | 2025-01-05 | shipped |
| SHP-000170 | ORD-000170 | DPD | TRK3560879 | 2025-01-05 | 2025-01-07 | shipped |
| SHP-000226 | ORD-000226 | DHL | TRK4629461 | 2025-01-07 | 2025-01-08 | shipped |

*Table 60.* Work Orders table for the Berlin Vintage Clothing Shop scenario (5 of 200 rows shown).

| WO ID | Customer ID | Date | Type | Status |
|-------|-------------|------|------|--------|
| WO-000001 | CUS-000001 | 2025-01-01 | alteration | completed |
| WO-000051 | CUS-000053 | 2025-01-02 | alteration | new |
| WO-000101 | CUS-000107 | 2025-01-04 | restoration | new |
| WO-000150 | CUS-000152 | 2025-01-05 | alteration | new |
| WO-000200 | CUS-000220 | 2025-01-07 | restoration | new |

## B.1. Location and Economic Context

Given a user prompt such as *"simulate a butcher shop in a morning market in Barcelona, Spain"*, the first LLM call extracts structured location data:

*Input prompt:* `"Extract the city, country, and currency from this business description.`
`Return ONLY valid JSON: {"city": "...", "country": "...", "currency": "..."}"`

*Output:* `{"city": "Barcelona", "country": "Spain", "currency": "EUR"}`

This call uses a low temperature (0.2) and a small token limit (128) since the task is purely extractive. If the LLM fails to produce valid JSON, sensible defaults (Berlin, Germany, EUR) are used.

The extracted country is matched against two bundled economic datasets:

- **GDP per capita** from the World Bank (2022–2024 columns; most recent non-null value), stored in `tablefactory/assets/gdp/GDP_by_country.csv`.

- **Corporate tax rate** from the Tax Foundation (statutory combined rate), stored in `tablefactory/assets/tax_rate/`.

For Spain, this yields GDP $\approx$ \$1.72 trillion (national) and a corporate tax rate of 25%. These values are injected verbatim into the scenario generation prompt (Step 2) so the LLM can calibrate prices and demand to the local economy. The same butcher shop prompt sent to a Swiss context would receive a GDP approximately $5\times$ higher, leading to correspondingly higher product prices and different supplier profiles.

### B.2. Scenario Generation Prompt

The core generation step uses a second LLM call with temperature 0.7 and a 16,384-token limit to accommodate large catalogs. The system prompt establishes the LLM's role as a *"simulation scenario designer for small-business ERP simulations"* and instructs it to be *"creative but realistic, prices should match the local economy."* The user prompt provides a detailed JSON schema requesting the following fields:

- **Product catalog** (15–100 items): each product includes name, category, list price, cost price, supplier assignment, initial stock, reorder point, and whether it requires a work order. The prompt specifies that 20–30% of products should require work orders (e.g., cutting, marinating, assembling).

- **Supplier directory** (2–4 suppliers): each with name, lead time (days), minimum order quantity (MOQ), and reliability score.

- **Staffing levels**: number and role of employees per entity type.

- **Entity role descriptions**: natural-language paragraphs that specialize each abstract entity to the business domain (see App. C for how these are consumed).

- **Payment methods**: locale-appropriate options (e.g., *Bizum* for Spain, *iDEAL* for the Netherlands).

- **Business policies**: return window (days), discount limits (%), escalation thresholds.

- **Demand model**: average daily customers and day-of-week traffic multipliers.

For the Barcelona butcher shop with GPT-5.4, this produces a store named *Carnisseria La Boquereta*, a 24-product catalog of locally named cuts (e.g., *Entrecot de ternera* at €29.50, *Butifarra fresca* at €12.50, *Secreto ibérico* at €18.90), four regional suppliers with Catalan county names (*Carns del Vallès*, *Porcs i Embotits del Penedès*), and a morning-weighted demand model peaking on Saturdays.

### B.3. Demand Model and Traffic Multipliers

The demand model includes per-weekday traffic multipliers, a dictionary mapping each day to a float that scales the base customer arrival rate. The prompt instructs the LLM to make these business-specific:

> *"daily_traffic_multipliers should reflect the business type. A bar should have higher Friday/Saturday traffic. A bakery should be steady on weekdays with a weekend peak. A morning market stall peaks on Saturday. Average across the week should be $\approx$1.0, range 0.4 (quiet) to 2.5 (peak)."*

For the Barcelona morning-market butcher shop, the LLM produces: Monday 0.82, Tuesday 0.90, Wednesday 0.96, Thursday 1.02, Friday 1.18, **Saturday 1.62**, Sunday 0.50. Saturday peaks reflect market-day demand; Sunday drops reflect reduced hours. These multipliers directly govern the number of Customer entities instantiated each simulated day, producing realistic weekly traffic patterns without hard-coded rules.

### B.4. LLM-Based Reward Model (RL Only)

A central challenge in applying RL to dynamically generated scenarios is reward function design: the relative importance of business events depends on the specific business domain, and hard-coding weights for every scenario is infeasible.

**Design rationale.** Two alternatives were considered: (1) use fixed default weights for all scenarios, which ignores domain-specific priorities; (2) query the LLM at every timestep to evaluate agent behavior, which is prohibitively expensive for RL training. The chosen approach is a middle ground: the LLM is consulted *once* during scenario generation to define per-entity reward weights, and the resulting weights are used deterministically throughout training. This preserves the speed of numeric reward computation while adapting the reward signal to the business domain.

**Generation process.** When the user selects an RL agent type (`ppo`, `sac`, or `ddpg`), the scenario generator issues an additional LLM call (1,024 tokens) with a structured prompt. The prompt provides the business type and store name, specifies per-entity reward fields with allowed ranges, and includes domain-specific guidance (e.g., *"For businesses where speed/freshness matters, increase operations and inventory weights. stockout_penalty should always be negative."*). The LLM returns a JSON dictionary of per-entity weights. Examples:

*Table 61.* Example LLM-generated reward weights for three business scenarios. Perishable businesses receive steeper stockout penalties; high-interaction businesses receive stronger sales conversion rewards.

| Scenario | Entity | Key Weight | Value |
|---|---|---|---|
| Butcher shop | Inventory | stockout_penalty | $-0.8$ |
| Butcher shop | Operations | work_order_completed | 1.5 |
| Butcher shop | Sales | order_converted | 1.0 |
| Electronics store | Inventory | stockout_penalty | $-0.3$ |
| Electronics store | Operations | work_order_completed | 0.8 |
| Cocktail bar | Sales | order_converted | 1.8 |
| Cocktail bar | Inventory | stockout_penalty | $-0.7$ |

**Reward formula.** Each entity's reward at timestep $t$ is a weighted sum of event counts:

$$r_t^i = \sum_k w_k^i \cdot N_k^i \tag{1}$$

where $N_k^i$ are event counts (orders completed, tickets closed, etc.) and $w_k^i$ are the LLM-generated weights. If the LLM fails to produce valid JSON, hardcoded defaults are used as fallback. This step is skipped entirely for LLM and rule-based agents, which do not use numeric rewards.

### B.5. Configuration Assembly

All LLM outputs are assembled into a single configuration dictionary with the same schema as a hand-written YAML file. The configuration includes two top-level blocks:

- **Environment**: entity definitions (with demand parameters, policies, and optional reward weights injected into each entity's `params`), product catalog, suppliers, staff, entity descriptions, payment methods, logistics defaults (shipping times, lost-package probability), and macroeconomic context.

- **Trainer**: algorithm selection (`customer_alg`, `sales_alg`, etc., all set to the chosen agent type), LLM model name and temperature, persona parameters, RL hyperparameters (learning rate, $\gamma$, buffer size, PPO epochs, clip epsilon, etc.), and logging settings.

The assembled config is saved as a YAML file (falling back to JSON if PyYAML is unavailable) alongside the simulation outputs. A concrete excerpt from a generated `business_config.json`:

```json
{
  "Environment": {
    "env_core": {
      "store_name": "Carnisseria La Boquereta",
      "episode_length": 365,
      "start_date": "2025-01-01"
    },
    "business_type": "butcher_shop",
    "user_prompt": "simulate a butcher shop in a morning
                    market in Barcelona, Spain",
    "location": {
      "city": "Barcelona",
      "country": "Spain",
      "currency": "EUR"
    },
    "macros": {
      "gdp_usd": 1722745978335.16,
      "corporate_tax_rate_pct": 25.0,
      "gdp_demand_multiplier": 1.0
    },
    "policies": {
      "return_window_days": 1,
      "sales_discount_limit_pct": 10,
      "escalation_threshold_refund_eur": 25
    }
  }
}
```

### B.6. Cross-Model Comparison

The same prompt produces different configurations from different LLMs. Tab. 62 compares GPT-5.4 and Claude Sonnet 4.6 on the Barcelona butcher shop prompt.

*Table 62.* Cross-model comparison on the Barcelona butcher shop prompt. Claude produces deeper Catalan immersion (Catalan product names, Catalan payment method labels) and a $2\times$ larger catalog.

| Aspect | GPT-5.4 | Claude Sonnet 4.6 |
|---|---|---|
| Store name | *Carnisseria La Boquereta* | *Carnisseria Can Rovira* |
| Products | 25 (Castilian names) | 55 (Catalan names) |
| Suppliers | 4 (1–2 day lead times) | 4 (mostly 1-day) |
| Avg customers/day | 46 | 55 |
| Payment methods | cash, credit, debit, mobile | Efectiu, Targeta, Bizum |

## C. Implementation Details of LLM-Based Agents

LLM agents translate entity observations into natural-language prompts, call an LLM API, and parse structured JSON responses back into numeric action vectors. This section details the prompt construction pipeline, multi-provider abstraction, response parsing, and text action forwarding.

## C.1. Multi-Provider LLM Wrapper

A unified `LLMWrapper` class provides a single `generate(model_name, prompt, system_prompt, ...)` interface that abstracts over four LLM providers. Provider detection is automatic: the wrapper inspects the model name string and routes to the appropriate backend. API clients are initialized lazily on first use.

*Table 63.* Supported LLM providers. Different providers produce different behavioral character in the generated data, same schema, different personality.

| Provider | Models | Detection | Behavioral Character |
|---|---|---|---|
| OpenAI | GPT-5.4, GPT-5.2, GPT-4o | `gpt` in name | Concise reasoning, clean JSON |
| Anthropic | Claude Sonnet 4.6, Opus 4.6 | `claude` in name | Empathetic, markdown fences |
| Google | Gemini 2.5 Flash | `gemini` in name | Fast, compact responses |
| DeepSeek | DeepSeek-R1 | `deepseek` in name | Detailed chain-of-thought |

## C.2. Prompt Construction

Each LLM agent call assembles a prompt from two layers of dynamically generated content:

**Layer 1: Scenario-derived context (static per simulation).** Read once from the generated config and reused across all days:

- Business type and location (e.g., *"Butcher shop / Carnicería in Barcelona, Spain (EUR)"*).

- Entity role description, a natural-language paragraph from the scenario generator that adapts each entity's persona to the business domain. For a butcher shop Sales entity: *"You are a sales assistant at a traditional Barcelona market butcher. You help customers choose cuts, handle complaints about freshness, and suggest recipes for seasonal products."*

- Product catalog, payment methods, and business policies.

**Layer 2: Runtime workload context (changes each day).** Assembled fresh on each simulated day from the entity's current state:

- **Customer**: shopping basket (product names, prices, quantities), cart total, pending status, available payment methods.

- **Sales**: current support ticket (ID, customer, issue type, channel, status, linked order).

- **Operations**: current work order (ID, type, status, materials).

- **Inventory**: product below reorder point (name, stock level, reorder point, target, cost) and its supplier (name, lead time, MOQ).

- **Manager**: return request (ID, refund amount, reason, escalation status).

Between the two layers, the agent's persona is injected under a `## Your Personality` heading (see App. E).

**Full Customer prompt example.** The following is an actual prompt sent to GPT-5.4 for an ENFP morning market shopper on Day 4 of the butcher shop simulation:

```
Neighborhood residents, market-goers, home cooks shopping
for fresh daily meals and weekend BBQs. You must decide
what to do RIGHT NOW.

## Your Personality
You are an ENFP customer: enthusiastic, people-focused,
imaginative, optimistic. You demonstrate high exploration,
low patience, very polite. Your tone is warm and friendly.
```

```
## Store
- Business: butcher_shop in Barcelona, Spain
- Currency: EUR

## Your Shopping Cart / Order
[{"name": "Costilla de cerdo", "unit_price": 9.80, "qty": 2},
 {"name": "Chorizo fresco", "unit_price": 11.80, "qty": 1}]
- Cart total: 31.40 EUR
- Returning customer (visited before): False

## Rules -- read carefully
1. Most customers (roughly 55-70%) COMPLETE a purchase.
2. About 20-30% place a DIRECT order; the rest add to cart.
3. Only ~5-10% open a pre-sales support ticket INSTEAD.
4. Accepted payment methods: cash, credit_card,
   debit_card, mobile_payment.

## Required Output -- return ONLY valid JSON
{"make_purchase": true, "direct_order": false,
 "payment_method": "cash", "open_ticket": false,
 "reasoning": "brief one-sentence explanation"}
```

### C.3. Per-Entity Observation Rendering

Each entity type receives a different prompt context based on its role. Tab. 64 summarizes what each entity observes and what decisions it outputs.

*Table 64.* Per-entity observation context and decision structure for LLM agents.

| Entity | Prompt Context (Dynamic) | Decision Output |
|---|---|---|
| Customer | Cart contents, total, payment methods | Purchase? Payment? Ticket? |
| Sales | Ticket ID, issue type, channel, order | Reply text, satisfaction, close? |
| Operations | Work order ID, type, status | Complete? Quality level (0–1)? |
| Inventory | Stock vs reorder point, supplier info | Create PO? Quantity factor? |
| Manager | Refund amount, reason, policies | Approve? Refund percentage? |

**Batch handling.** The Customer entity produces a variable number of items per day (one per arriving customer). LLM agents process each customer individually, issuing one API call per customer with that customer's specific basket and persona. This preserves individual decision context at the cost of higher API usage. A 7-day simulation with ∼50 customers per day requires ∼350 Customer API calls plus ∼7 calls each for the four staff entities.

### C.4. Response Parsing and Fallbacks

The raw LLM response text is parsed through a four-stage cascade:

1. Direct `json.loads(text)`, works for models that return clean JSON (typically GPT-5.4).

2. Extract content from a ```` ```json ...    ``` ```` fenced block, handles Claude's markdown wrapping.

3. Extract content from any ```` ``` ...    ``` ```` fenced block.

4. Extract the substring between the first { and last }, catches models that add preamble before JSON.

If all stages fail, or if the parsed JSON lacks required fields, the agent falls back to hardcoded default action vectors: Customer $[0.6, 0.3, 0.0, 0.1]$, Sales $[0.7, 0.5, 0.3]$, Operations $[1.0, 0.8]$, Inventory $[1.0, 0.5]$, Manager $[0.5, 0.8]$. This guarantees the simulation never crashes regardless of LLM output quality.

The per-entity parser maps JSON fields to a numeric action vector. For the Customer entity:

*Table 65.* JSON-to-action-vector mapping for the Customer entity. The resulting vector $[1.0, 0.0, 0.66, 0.0]$ is processed by the same entity threshold logic used for RL agents.

| JSON Field | Mapping | Value |
|---|---|---|
| `"make_purchase": true` | true→1.0, false→0.0 | 1.0 |
| `"direct_order": false` | true→1.0, false→0.0 | 0.0 |
| `"payment_method": "mobile"` | credit→0.0, paypal→0.33, bank→0.66 | 0.66 |
| `"open_ticket": false` | true→1.0, false→0.0 | 0.0 |

## C.5. Text Action Forwarding

LLM agents produce richer outputs than numeric actions alone. For the Sales entity, the LLM generates a natural-language reply message alongside its action decisions. The runner collects these structured decisions via `_last_decisions` and forwards them to the Sales entity before the environment step. The entity records the reply text in the `ticket_messages` table of the WorldState, enriching the output data with realistic conversational content. Example ticket replies from an actual butcher shop run:

> **Ticket TIC-000001** (delivery inquiry for order ORD-000040):
> *"Hello, regarding order ORD-000040: I'm checking the current delivery status with our carrier now. Please reply with your full delivery address and any tracking email you received so I can confirm the shipment and update you quickly."*

> **Ticket TIC-000003** (warranty claim for order ORD-000071):
> *"Hi. For warranty on order ORD-000071, please send 3 photos/video of the issue, the serial number, and confirm the purchase date. Once received, we'll verify coverage and give you the next step immediately."*

These text outputs do not affect simulation dynamics but enrich the generated dataset for downstream NLP tasks.

# D. Implementation Details of RL-Based Agents

All RL agents share the same network backbone and training pipeline, differing only in their optimization objective and update rule. This section provides full algorithmic details.

## D.1. Network Architecture

All RL agents use two-hidden-layer MLPs with 256 units and ReLU activations. Because observation dimensions vary across scenarios (entity-specific features differ, and persona traits are optionally appended), networks are **constructed lazily**: weights are initialized to `None` and allocated on the first call to `get_action`, once the true input dimension is known. This avoids hard-coding observation sizes and allows the same agent code to serve any dynamically generated scenario.

```
obs (11-14 dims + 10 persona) -> Linear(256) -> ReLU
                              -> Linear(256) -> ReLU
                              -> action (2-4 dims)
```

## D.2. PPO (On-Policy)

PPO optimizes a clipped surrogate objective using Generalized Advantage Estimation (GAE). A shared MLP torso branches into a *policy head* (outputs action means, with a learnable log-std parameter) and a *value head* (outputs a scalar state-value estimate). The objective is:

$$\mathcal{L}^{\text{PPO}} = \mathbb{E}\Big[\min\Big(\rho_t \hat{A}_t, \ \text{clip}(\rho_t, 1-\epsilon, 1+\epsilon)\,\hat{A}_t\Big)\Big] + c_v \mathcal{L}^{\text{value}} - c_e H[\pi] \tag{2}$$

where $\rho_t = \pi_\theta(a_t|s_t)/\pi_{\theta_{\text{old}}}(a_t|s_t)$ is the importance ratio, $\hat{A}_t$ is the GAE advantage with $\lambda = 0.95$, $\mathcal{L}^{\text{value}}$ is the mean-squared value error, and $H[\pi]$ is the entropy bonus. After each simulated day, all transitions collected that day are used for 4 mini-batch epochs, then discarded (on-policy).

## D.3. SAC (Off-Policy)

SAC maximizes a maximum-entropy objective using twin Q-networks to mitigate overestimation bias. Two independent Q-networks $Q_{\phi_1}, Q_{\phi_2}$ minimize the soft Bellman error against target networks:

$$\mathcal{L}^Q = \mathbb{E}\left[(Q_\phi(s,a) - y)^2\right], \quad y = r + \gamma \left(\min_j \bar{Q}_{\phi_j}(s',a') - \alpha \log \pi(a'|s')\right) \tag{3}$$

The actor maximizes $\mathbb{E}\left[\min_j Q_{\phi_j}(s,\tilde{a}) - \alpha \log \pi(\tilde{a}|s)\right]$ with $\tilde{a}$ sampled via the reparameterization trick (Gaussian policy, squashed through $\tanh$). The temperature $\alpha$ is automatically adjusted to maintain a target entropy of $-\dim(\mathcal{A})$. After each simulated day, one gradient step is performed on a mini-batch sampled from the replay buffer.

## D.4. DDPG (Off-Policy)

DDPG learns a deterministic policy $\mu_\theta(s)$ with additive exploration noise. A single Q-network $Q_\phi(s,a)$ minimizes MSE against a slowly-tracking target network:

$$\mathcal{L}^Q = \mathbb{E}\left[(Q_\phi(s,a) - (r + \gamma \bar{Q}_{\bar{\phi}}(s', \mu_{\bar{\theta}}(s'))))^2\right] \tag{4}$$

The actor maximizes $\mathbb{E}[Q_\phi(s, \mu_\theta(s))]$. Additive Gaussian noise $\mathcal{N}(0, \sigma^2)$ with $\sigma = 0.1$ is added to the actor output during data collection and removed at evaluation.

## D.5. Action Rescaling

RL networks output raw actions $\tilde{a}_t^i \in [-1, 1]^{d_i}$ via $\tanh$ activation. The runner applies centralized rescaling before passing actions to entities:

$$a_t^i = \frac{\tilde{a}_t^i + 1}{2} \in [0, 1]^{d_i} \tag{5}$$

**Worked example.** An ENFP morning shopper has a €31.40 basket. The PPO network outputs $[0.2, -0.6, 0.1, -0.8]$:

```
Network:  [0.2, -0.6,  0.1, -0.8]
Runner:   (action + 1) / 2 = [0.60, 0.20, 0.55, 0.10]

  action[0] = 0.60 > 0.5  -> PURCHASE (the shopper buys)
  action[1] = 0.20 <= 0.5 -> add to cart first
  action[2] = 0.55 * 3 = 1.65 -> int = 1 -> paypal
  action[3] = 0.10 <= 0.5 -> do NOT open ticket
```

This produces identical WorldState mutations as an LLM agent returning {`"make_purchase"`: `true`, `"direct_order"`: `false`, `"payment_method"`: `"paypal"`, `"open_ticket"`: `false`}.

## D.6. Hyperparameter Summary

Tab. 66 consolidates all RL hyperparameters.

## D.7. Training Pipeline and Logging

Training proceeds in daily epochs. The runner collects transitions $(o_t, a_t, r_t, o_{t+1}, d_t)$ into per-agent buffers. PPO consumes its buffer on-policy at the end of each day, while SAC and DDPG sample from a persistent replay buffer. For the Customer entity, batched observations are mean-pooled before storage (see App. F), ensuring consistent tensor shapes across days with varying customer counts.

Logged metrics via TensorBoard include: actor loss, critic/value loss, entropy (SAC), average and total rewards per entity, and episode counts. Training curves and model checkpoints are saved to the output directory for offline analysis.

*Table 66.* RL hyperparameters for all three algorithms.

| Parameter | PPO | SAC | DDPG |
|---|---|---|---|
| Learning rate | $3 \times 10^{-4}$ | $3 \times 10^{-4}$ | $1 \times 10^{-3}$ |
| Batch size | 64 | 256 | 64 |
| Discount ($\gamma$) | 0.99 | 0.99 | 0.99 |
| Replay buffer | — | $10^6$ | $10^6$ |
| Target update ($\tau$) | — | 0.005 | 0.005 |
| Entropy coefficient | 0.01 | auto | — |
| Exploration noise ($\sigma$) | — | — | 0.1 |
| GAE $\lambda$ | 0.95 | — | — |
| Clip $\epsilon$ | 0.2 | — | — |
| Value coefficient ($c_v$) | 0.5 | — | — |
| Max gradient norm | 0.5 | — | — |
| Mini-batch epochs/day | 4 | 1 | 1 |

# E. Implementation Details of MBTI Persona Layer

The persona system introduces behavioral diversity by assigning each agent a personality profile grounded in the Myers–Briggs Type Indicator (MBTI) framework. The same profile drives both LLM and RL agents through different channels.

## E.1. MBTI Sampling

Four primary dimensions are sampled independently from a symmetric Beta distribution:

$$E, S, T, J \sim \text{Beta}(\alpha, \alpha), \quad \alpha = 2.0 \tag{6}$$

with complements $I = 1 - E$, $N = 1 - S$, $F = 1 - T$, $P = 1 - J$. The Beta(2,2) distribution is unimodal at 0.5, favouring moderate personalities while still allowing extreme types. The four-letter MBTI code is derived by thresholding each axis at 0.5 (e.g., $E > 0.5 \Rightarrow$ E, else I).

**Worked example.** For customer #7 (seed = 42 + 7):

*Table 67.* MBTI dimension sampling for customer #7 (seed = 42 + 7), yielding the ENFP personality type.

| Dimension Pair | Sampled Value | Complement | Winner |
|---|---|---|---|
| E / I | $E = 0.72$ | $I = 0.28$ | **E** |
| S / N | $S = 0.31$ | $N = 0.69$ | **N** |
| T / F | $T = 0.38$ | $F = 0.62$ | **F** |
| J / P | $J = 0.25$ | $P = 0.75$ | **P** |

Result: **ENFP** $\rightarrow$ adjectives: *enthusiastic, people-focused, imaginative, optimistic*.

## E.2. Behavioral Trait Derivation

The MBTI dimensions are mapped to 10 continuous behavioral traits via linear combinations with small Gaussian noise ($\sigma = 0.03$), clipped to $[0, 1]$:

$$\text{exploration} = 0.5 + 0.3N + 0.2E + 0.2P - 0.2S - 0.1I - 0.2J \tag{7}$$

$$\text{patience} = 0.5 + 0.25I + 0.25J - 0.15E - 0.15P \tag{8}$$

$$\text{price\_sensitivity} = 0.5 + 0.15S + 0.15T - 0.15N - 0.15F \tag{9}$$

$$\text{brand\_loyalty} = 0.5 + 0.3J + 0.2F - 0.3P \tag{10}$$

$$\text{politeness} = 0.5 + 0.3F + 0.2J - 0.2T - 0.1P \tag{11}$$

$$\text{empathy} = 0.5 + 0.4F - 0.4T \tag{12}$$

$$\text{rule\_rigidity} = 0.5 + 0.3J + 0.2S - 0.3P - 0.2N \tag{13}$$

$$\text{upsell\_tendency} = 0.5 + 0.2E + 0.2F + 0.1P \tag{14}$$

$$\text{complaint\_freq} = 0.3 + 0.2T + 0.2E - 0.3 \cdot \text{patience} \tag{15}$$

$$\text{support\_contact} = 0.4 + 0.3E + 0.2 \cdot \text{complaint\_freq} - 0.2 \cdot \text{patience} \tag{16}$$

**Worked examples.** For the ENFP customer above ($E=0.72, N=0.69, F=0.62, P=0.75$): exploration $\approx 0.86$, patience $\approx 0.33$, price sensitivity $\approx 0.41$, politeness $\approx 0.66$. This customer tries unfamiliar cuts (*Conejo troceado*), leaves quickly if the queue is long, and is friendly at the counter.

For an ISTJ customer ($I=0.80, S=0.70, T=0.60, J=0.80$): exploration $\approx 0.22$, patience $\approx 0.73$, rule rigidity $\approx 0.78$. This customer buys familiar staples (*Carne picada*, *Costilla de cerdo*), waits patiently, and expects predictable service.

### E.3. Communication Style

A communication style is derived from the MBTI dimensions for use in LLM prompts:

- **Verbosity**: $v = 0.33E + 0.33N - 0.34I + 0.33$; mapped to *low* ($v < 0.33$), *medium*, or *high* ($v \geq 0.67$).

- **Tone**: $F > 0.6 \Rightarrow$ *warm\_friendly*; $T > 0.6 \wedge J > 0.6 \Rightarrow$ *neutral\_professional*; $E > 0.6 \wedge T > 0.6 \Rightarrow$ *direct*; else *neutral\_professional*.

- **Emotive intensity**: $0.5 + 0.3F + 0.2E$, clipped to $[0, 1]$.

- **Directness**: $0.5 + 0.3T + 0.2E - 0.1F$, clipped to $[0, 1]$.

### E.4. LLM Prompt Seed

The persona data is assembled into a natural-language description used as a prompt prefix for LLM agents. For the ENFP customer:

*"You are an ENFP customer: enthusiastic, imaginative, warm, perceptive. You demonstrate high exploration, low patience, very polite. You tend to be expressive and detailed in communication. Your tone is warm and friendly."*

An ISTJ sales agent would receive: *"You are an ISTJ sales agent: responsible, thorough, dependable, practical. You demonstrate high rule rigidity, high patience. You communicate concisely. Your tone is neutral and professional."*

### E.5. Dual Consumption: LLM vs. RL

The same persona profile is consumed through two channels depending on the agent type:

- **LLM agents** receive the prompt seed text as a natural-language prefix. The LLM's responses naturally reflect the described personality, an "impatient" customer is more likely to complain; a "rule-rigid" manager is less likely to approve borderline refunds.

- **RL agents** receive the 10 trait values as a float vector appended to the observation: $o_t^{\text{persona}} = [\text{patience}, \text{price\_sens}, \ldots, \text{upsell}] \in [0, 1]^{10}$. The policy network learns persona-conditioned behavior from these features.

**Persona assignment.** Staff entities (Sales, Operations, Inventory, Manager) receive a **single persistent persona** for the entire simulation, seeded by `seed + hash(entity_name)`. Customer entities generate a **unique persona per customer**, seeded by `seed + hash(customer_id)`, ensuring that the same customer ID always produces the same personality across runs.

## E.6. Behavioral Impact Example

Two shoppers on the same morning in the Barcelona butcher shop, with the same product catalog available:

*Table 68.* Behavioral differences between two persona types shopping at the same butcher shop on the same morning.

|  | **Shopper #7 (ENFP)** | **Shopper #12 (ISTJ)** |
|---|---|---|
| Basket | *Conejo troceado + Pinchos morunos* | *Carne picada + Costilla* |
| Basket style | Adventurous, varied categories | Conservative, familiar staples |
| Ticket? | No (too impatient to wait) | No (patient, no complaints) |
| Payment | mobile_payment (spontaneous) | cash (habitual) |

# F. Observation and Action Design

This section describes how entity states are represented as observations and how agent decisions are structured as actions. The same underlying entity state is consumed differently by LLM and RL agents.

## F.1. General Structure

Every entity's observation consists of two components:

$$o_t^{\text{base}} = [o_t^{\text{global}} ; o_t^{\text{entity}}] \tag{17}$$

where $o_t^{\text{global}} \in \mathbb{R}^7$ captures system-wide state and $o_t^{\text{entity}} \in \mathbb{R}^{3-4}$ captures entity-specific features. All features are normalized to $[0, 1]$: continuous values are scaled by expected maximums, and binary flags remain discrete. For RL agents, 10 persona trait floats are appended:

$$o_t^i = [o_t^{\text{global}} ; o_t^{\text{entity}} ; o_t^{\text{persona}}] \in \mathbb{R}^{d_o^i} \tag{18}$$

Action spaces are continuous and fixed per entity type: Customer ($d=4$), Sales ($d=3$), Operations ($d=2$), Inventory ($d=2$), Manager ($d=2$). Entities interpret actions via **threshold logic**: a dimension value $a_k > 0.5$ triggers a binary decision (e.g., "purchase" or "approve"), while continuous dimensions (e.g., refund percentage, quantity factor) are used as-is.

## F.2. Global Observation Vector

The 7-dimensional global observation vector is shared by all entities. Tab. 69 details each dimension with normalization and example values from Day 4 of the butcher shop.

*Table 69.* Global observation vector (7 dimensions), shared by all entities.

| Dim | Feature | Normalization | Example Raw | Normalized |
|---|---|---|---|---|
| 0 | Total customers | $\min(n/1000, 1.0)$ | 46 customers | 0.046 |
| 1 | Total orders | $\min(n/500, 1.0)$ | 38 orders | 0.076 |
| 2 | Open tickets | $\min(n/50, 1.0)$ | 2 tickets | 0.040 |
| 3 | Pending work orders | $\min(n/20, 1.0)$ | 5 work orders | 0.250 |
| 4 | Total stock | $\min(n/500, 1.0)$ | 312 units | 0.624 |
| 5 | Low-stock count | $\min(n/10, 1.0)$ | 3 SKUs below 5 | 0.300 |
| 6 | Simulation progress | step/episode_length | Day 4 of 365 | 0.011 |

## F.3. Per-Entity Local Features

Each entity observes only what is relevant to its role. Tab. 70 details the per-entity local observation features.

*Table 70.* Per-entity local observation features and action dimensions.

| Entity | Local Dims | Key Features | Act Dims |
|---|---|---|---|
| Customer | 4 | Basket total/10000, is pending, seed factor, is B2B | 4 |
| Sales | 4 | Is pre-sales, has linked order, channel, ticket ID | 3 |
| Operations | 3 | Is custom request, is in progress, work order ID | 2 |
| Inventory | 4 | On-hand/100, reorder point/100, target/100, cost/1000 | 2 |
| Manager | 3 | Refund amount/1000, exceeds escalation, return ID | 2 |

**Customer batch handling.** When multiple shoppers arrive (5–20 per morning), each produces a separate observation row. RL agents **mean-pool** the $N \times d$ matrix into a single $1 \times d$ summary vector, forward it through the policy network once, and **tile** the resulting action across all $N$ customers. This ensures fixed tensor shapes for gradient computation regardless of daily customer volume. Staff entities always produce a single observation, so no pooling is needed. LLM agents handle each customer individually (see App. C).

## F.4. Action Space Design

Actions are continuous in $[0, 1]$ and interpreted via entity-specific threshold logic. For the Customer entity (4 dimensions):

*Table 71.* Customer entity action space (4 dimensions).

| Dim | Action | Interpretation |
|---|---|---|
| 0 | Make purchase | $> 0.5 \rightarrow$ buy; $\leq 0.5 \rightarrow$ leave without purchasing |
| 1 | Direct order | $> 0.5 \rightarrow$ place order directly; $\leq 0.5 \rightarrow$ add to cart first |
| 2 | Payment method | Continuous $\rightarrow \lfloor \text{val} \times 3 \rfloor$: 0=credit, 1=paypal, 2=bank |
| 3 | Open ticket | $> 0.5 \rightarrow$ open a pre-sales support ticket instead of buying |

## F.5. MDP Formulation and Rewards

The simulation is modeled as a decentralized multi-agent Markov Decision Process over discrete timesteps, where each timestep $t$ represents one simulated day. For each entity $i \in \{\text{cust}, \text{sales}, \text{ops}, \text{inv}, \text{mgr}\}$:

$$o_t^i \in \mathbb{R}^{d_o^i} \qquad \text{(observation)} \tag{19}$$

$$a_t^i \in [0, 1]^{d_a^i} \qquad \text{(continuous action)} \tag{20}$$

$$r_t^i = \sum_k w_k^i \cdot N_k^i \qquad \text{(weighted event counts)} \tag{21}$$

Each agent maximizes its own expected discounted return $J(\pi^i) = \mathbb{E}[\sum_{t=0}^{T} \gamma^t r_t^i]$ with $\gamma = 0.99$. The per-entity reward functions are:

$$r_t^{\text{cust}} = w_1 \cdot N_{\text{orders}} + w_2 \cdot N_{\text{carts}} + w_3 \cdot N_{\text{tickets}} \tag{22}$$

$$r_t^{\text{sales}} = w_1 \cdot N_{\text{conversions}} + w_2 \cdot N_{\text{handled}} + w_3 \cdot N_{\text{closed}} \tag{23}$$

$$r_t^{\text{ops}} = w_1 \cdot N_{\text{work\_orders\_completed}} \tag{24}$$

$$r_t^{\text{inv}} = w_1 \cdot N_{\text{purchase\_orders}} + w_2 \cdot N_{\text{stockouts}} \tag{25}$$

$$r_t^{\text{mgr}} = w_1 \cdot N_{\text{returns\_processed}} + w_2 \cdot N_{\text{returns\_approved}} \tag{26}$$

The weights $w_k^i$ are scenario-specific. When using RL agents with dynamic scenario generation, they are produced by the LLM-based reward model (App. B.4). For LLM agents or static configs, sensible defaults are used (e.g., $w_1{=}1.0, w_2{=}0.3, w_3{=}0.1$ for Customer).

## F.6. Summary Table

Tab. 72 consolidates all per-entity MDP elements.

*Table 72.* Per-entity MDP elements. All observations include the shared 7-D global state vector. For RL agents, 10 persona trait floats are appended. Actions are continuous in $[0, 1]$ and interpreted via thresholds.

| Entity | Observation $o_t$ | Action $a_t \in [0, 1]$ | Reward Signals |
|---|---|---|---|
| Customer | Basket value, pending, seed, B2B | Purchase, direct, payment, ticket | Orders, carts, tickets |
| Sales | Ticket type, order, channel, rand | Satisfaction, convert, close | Conversions, handled, closed |
| Operations | Custom flag, in-progress, rand | Complete work, quality | Completed work orders |
| Inventory | Stock, reorder pt, target, cost | Create PO, quantity factor | POs created, stockout penalty |
| Manager | Return value, escalation, rand | Approve refund, refund % | Returns processed, approved |

## G. Entity Layer Design

### G.1. BaseEntity Interface

All five entities inherit from a common `BaseEntity` abstract base class with five methods:

- `prepare_batch(world_state, config, ...)`, set up the entity's workload for the current day (e.g., sample new customers, fetch open tickets, find low-stock products).

- `get_observation()`, produce a numeric observation vector from the current workload.

- `get_action(actions)` / `step(world_state, t)`, receive the agent's action and execute business logic against the shared WorldState.

- `get_reward()`, compute a scalar reward from configurable weights and event counts.

- `get_metrics()`, report per-day statistics for logging.

This interface makes entities fully agent-agnostic: the same Customer entity executes identically whether driven by an LLM, a PPO policy, or a handcrafted rule.

### G.2. Five Entity Types

The five abstract roles cover the core operations of a small-business ERP:

**Customer.** Each simulated day, a stochastic number of new customers arrive (governed by day-of-week traffic multipliers from the demand model). Each customer receives a shopping basket sampled from the product catalog. The action space (4 dimensions) determines whether each customer makes a purchase, places a direct order or uses a cart, selects a payment method, and opens a pre-sale support ticket. Post-sale events, shipments, inventory deductions, work orders for products requiring preparation, after-sale support tickets, and return requests, are generated automatically based on configurable probabilities.

**Sales.** Processes open support tickets from the WorldState. Actions (3 dimensions) control customer satisfaction scoring, cart-to-order conversion, and ticket closure. When driven by an LLM agent, the entity also records natural-language reply messages forwarded by the runner.

**Operations.** Handles pending work orders. The same entity code processes "butcher a pork loin" in a butcher shop and "assemble a gaming PC" in a computer store, the work order type and material list come from the config. Actions (2 dimensions) determine whether to complete a work order and at what quality level, triggering material deductions from inventory.

**Inventory.** Monitors products with stock below the reorder point. Actions (2 dimensions) decide whether to create a purchase order and in what quantity factor ($0.5\times$–$1.5\times$ the target quantity). Orders arrive from suppliers after a configurable lead time.

**Manager.** Processes unresolved return requests. Actions (2 dimensions) determine whether to approve a refund and at what percentage (0–100%).

### G.3. Environment (`ShopEnvironment`)

The environment manages the shared WorldState, an in-memory relational database, and coordinates five entity instances through a standard `reset/step` interface. On each simulated day the environment:

1. Prepares entity workloads: samples new customers (Customer), fetches open tickets (Sales), fetches pending work orders (Operations), identifies low-stock products (Inventory), and fetches unresolved returns (Manager).

2. Assembles observations: a shared 7-D global state vector concatenated with entity-specific features.

3. Dispatches actions to entities, which execute business logic against the WorldState.

4. Advances the simulation clock and receives due purchase orders from suppliers (restocking inventory).

The environment is agnostic to both the business domain and the agent algorithm: it reads all domain-specific parameters (catalog, suppliers, policies, demand model) from the configuration.

## H. Agent Interactions

### H.1. Decentralized Coupling Through WorldState

Entities are decentralized: they share no global reward and do not communicate directly. All interactions arise implicitly through the WorldState, one agent's output becomes another agent's input on the next step. Tab. 73 enumerates the interaction pathways.

*Table 73.* Inter-entity interaction pathways, all mediated through the shared WorldState.

| Interaction | Trigger | WorldState Effect |
|---|---|---|
| Customer → Sales | Order or inquiry | Creates support ticket |
| Customer → Inventory | Fulfilled order | Depletes stock quantities |
| Customer → Operations | Product needs prep | Generates work order |
| Customer → Manager | Post-sale return | Populates return queue |
| Operations → Inventory | Work order completed | Consumes materials (stock adj.) |
| Inventory → Suppliers | Stock below reorder | Creates purchase order |
| Sales → Customer | Ticket resolved / converted | Closes the feedback loop |

No agent reads another agent's internal state. The only shared structure is the WorldState, a collection of mutable tables that each agent can read from and write to during its step.

### H.2. LLM Text Enrichment

When LLM agents are used, the implicit couplings are augmented with natural-language artifacts:

- Sales agents write ticket **reply messages** → stored in the `ticket_messages` table.

- All agents produce **reasoning traces** explaining their decisions → stored in the `actions_log` table.

These text outputs do **not** affect simulation dynamics but enrich the generated dataset for downstream NLP tasks. Rule-based and RL agents produce only numeric actions, so these tables remain empty or contain only structured metadata for non-LLM runs.

## I. Output Schema

### I.1. Table Categories

Each simulation run produces a set of CSV tables organized into five groups, as shown in Tab. 74.

*Table 74.* Output table categories. The exact number of tables may vary depending on the generated business configuration.

| Category | Tables |
|---|---|
| Master data | `customers`, `products`, `suppliers`, `staff` |
| Commerce | `carts`, `cart_items`, `orders`, `order_items`, `payments`, `shipments`, `returns` |
| Inventory & procurement | `inventory_snapshots`, `purchase_orders`, `purchase_order_items`, `stock_adjustments` |
| Customer support | `support_tickets`, `ticket_messages` |
| Operations | `work_orders`, `work_order_items`, `actions_log` |

## I.2. Causal Data Chain Example

The following traces one customer through the full pipeline, using real output rows from the Barcelona butcher shop run (GPT-5.4, 7 days):

**Customer record:** `CUS-000004, Consumer, ``126 Main St, Barcelona'', 2025-01-01`

**Order:** `ORD-000004, CUS-000004, 2025-01-01, shipped, credit_card`

**Order items:**

- `ORD-000004, PORK05 (Pinchos morunos de cerdo), qty 2, €13.90`

- `ORD-000004, READY01 (Ready-to-cook item), qty 2, €13.80`

- `ORD-000004, BEEF02 (Entrecot de ternera), qty 2, €29.50`

**Payment:** `PAY-000004, ORD-000004, €114.40, credit_card, captured`

**Shipment:** `SHP-000004, ORD-000004, DPD, TRK2402747, shipped 2025-01-01, est. delivery 2025-01-03`

**Downstream effects:** The order for Pinchos morunos (which requires marinating) spawns a work order for the Operations entity; the fulfilled order depletes inventory, potentially triggering a purchase order from the Inventory entity. Every row is causally linked by simulated behavior.

## I.3. Additional Artifacts

Beyond CSV tables, each run exports:

- **Configuration snapshot**: the full `business_config.json` (or YAML) used for the run.

- **Per-day metrics history**: `metrics_history.json` with daily totals for orders, revenue, tickets, work orders, etc.

- **Run metadata**: `run_metadata.json` with agent type, model name, seed, number of epochs, and runtime.

- **Persona profiles**: per-agent JSON files in a `personas/` subdirectory (when the persona system is enabled).

- **RL artifacts** (when applicable): trained model weights, TensorBoard logs, and training curves.

# J. Code Organization

```
tablefactory/
  main.py                  # CLI entry point (--user_prompt, --agent_type)
  scenario_generator.py    # Free-form prompt -> config via LLM
  runner.py                # Simulation loop, training, export
```

```
world_state.py               # In-memory relational database
exporter.py                  # WorldState -> CSV/JSON export
env/
  env_core.py                # ShopEnvironment (reset/step interface)
  set_observation.py         # Global + entity observation assembly
entities/
  base.py                    # BaseEntity ABC
  customer.py                # Customer entity (baskets, orders, post-sale)
  sales.py                   # Sales entity (ticket handling, conversion)
  operations.py              # Operations entity (work order completion)
  inventory.py               # Inventory entity (reorder, PO creation)
  manager.py                 # Manager entity (return approval)
agents/
  base.py                    # BaseAgent ABC, persona augmentation
  econgym_wrappers.py        # Unified agent interface adapter
  llm/
    llm_agent.py             # LLM-based agent (prompt -> API -> action)
    llm_wrapper.py           # Multi-provider LLM abstraction
    prompts.py               # Parameterized prompt templates
  rl/
    models.py                # Actor-Critic, Q-network architectures
    ppo_agent.py             # PPO agent
    sac_agent.py             # SAC agent
    ddpg_agent.py            # DDPG agent
  rule_based/
    rules_core.py            # Deterministic heuristic baseline
personas/
  persona_generator.py       # MBTI sampling -> traits -> prompt seeds
config/
  *.yaml                     # Static and generated scenario configs
assets/
  gdp/, tax_rate/            # Economic data for scenario generation
```

