# OpenReview forum: "TableFactory: Generating Semantically Linked Tabular Data via Multi-Agent Behavioral Simulation"
_ICML.cc/2026/Workshop/FMSD — FMSD @ ICML 2026 Poster_

### Official Review · Reviewer_DEU2 · 2026-05-19
**Interesting simulation framework undercut by insufficient empirical validation**

**Rating:** 4
**Confidence:** 3

**Review:**

Summary: This paper introduces TableFactory, a prompt-conditioned framework for generating semantically linked relational tabular data through multi-agent behavioral simulation rather than by directly fitting single-table distributions. Given a short business prompt, the system uses an LLM to create a business configuration with products, suppliers, demand patterns, policies, macroeconomic context, and role descriptions. Then, five abstract entities interact through a shared relational world state. These entities can be controlled by LLM agents, RL agents, or rule-based policies, with an MBTI-inspired persona layer intended to add behavioral diversity. The system exports multi-table CSV bundles with foreign-key links, metadata, configurations, and logs. The paper demonstrates the approach on small-business scenarios and reports diagnostic comparisons across LLM backends, a PPO controller, and a rule-based controller.

Strong Points:

S1: The paper targets the scarcity of realistic multi-table operational data, which is relevant to structured-data modeling.

S2: Instead of treating tabular synthesis as distribution matching over an observed table, the paper frames data generation as simulating the behavioral processes that create records.

S3: The decomposition into scenario generation, entity roles, agent controllers, personas, a shared world state, and an export layer is easy to understand.


Potential Weaknesses:

W1: The MBTI-inspired persona layer seems interesting but is not empirically validated. The paper does not show that the sampled personas produce statistically meaningful behavioral differences, improve dataset utility, or correspond to real customer variation.

W2: The paper claims high-fidelity and semantically linked data generation, but the experiments mostly report internal counts and ablations. There is no quantitative validation of utility, fidelity, or the downstream performance of the generated data.

W3: In addition, the paper does not compare against existing tabular or relational synthetic-data generators.

---

### Official Review · Reviewer_W3Rr · 2026-05-22
**A multi-agent simulator designed to generate semantically linked, multi-table operational data**

**Rating:** 8
**Confidence:** 4

**Review:**

Summary:

This work introduces TableFactory, a multi-agent simulator designed to generate semantically linked, multi-table operational data, rather than relying on single-table distribution matching.

Strength:

1. A timely data generation framework shows practically value when it comes to multi-table generation, utilizing behavioral simulation. The framework provides guarantee across complex data schemas.
2. The architecture provides rich contextual grounding such as macroeconomic variables and MBTI personality, etc.

Area for Improvements:

1. The idea of the framework is fascinating and naturally introduces a challenging evaluating setup. The current preliminary evaluation framework lacks quantitative comparisons. Although it might be difficult to judge the quality of the outputs, maybe applying synthetic data and real data to appropriate downstream applications and measure performance could give us quality indicators.
2. This work could benefit more from having a cost analysis. As the scale of the schema goes up, the cost (in terms of price and time) of training, simulating and generating will go up. Deriving the cost model and understand if the cost is linear would help other use cases.